



# 1 Volcanic SO₂ Layer Height by TROPOMI/S5P; validation against

# 2 IASI/MetOp and CALIOP/CALIPSO observations.

Maria-Elissavet Koukouli[1], Konstantinos Michailidis[1], Pascal Hedelt[2], Isabelle A. Taylor[3], Antje Inness[4],
Lieven Clarisse[5], Dimitris Balis[1], Dmitry Efremenko[2], Diego Loyola[2], Roy G. Grainger[3] and Christian
Retscher[6]
[1] Laboratory of Atmospheric Physics, Aristotle University of Thessaloniki, Greece.
[2] German Aerospace Center (DLR), Remote Sensing Technology Institute, Oberpfaffenhofen, Germany.
[3] COMET, Sub-department of Atmospheric, Oceanic and Planetary Physics, University of Oxford, UK.
[4] European Centre for Medium-Range Weather Forecasts (ECMWF), Reading, UK.
[5] Université Libre de Bruxelles (ULB), Spectroscopy, Quantum Chemistry and Atmospheric Remote Sensing (SQUARES),
Brussels, Belgium.
[6] European Space Agency, ESRIN, Frascati, Rome.
*Corresponding author*: Hedelt, Pascal Andre, Pascal.Hedelt@dlr.de
**Abstract.** Volcanic eruptions eject large amounts of ash and trace gases such as sulphur dioxide (SO₂) into the atmosphere. A
significant difficulty in mitigating the impact of volcanic SO₂ clouds on air traffic safety is that these gas emissions can be
rapidly transported over long distances. The use of space-borne instruments enables the global monitoring of volcanic SO₂
emissions in an economical and risk-free manner. Within the European Space Agency (ESA) Sentinel-5p+ Innovation project,
the S5P SO₂ Layer Height (S5P+I: SO₂ LH) activities led to the improvements on the retrieval algorithm and generation of the
corresponding near-real-time S5P SO₂ LH products. These are currently operationally provided, in near-real-time, by the
German Aerospace Center (DLR) in the framework of the Innovative Products for Analyses of Atmospheric Composition,
INPULS, project. The main aim of this paper is to present its extensive verification, accomplished within the S5P+I: SO₂ LH
project, over major recent volcanic eruptions, against collocated space-born measurements from the IASI/Metop and
CALIOP/CALIPSO instruments, as well as assess its impact on the forecasts provided by the Copernicus Atmospheric
Monitoring Service, CAMS. The mean difference between S5P and IASI observations for the Raikoke 2019, the Nishinoshima
2020 and the La Soufrière-St Vincent, 2021 eruptive periods is ~0.5±3km, while for the Taal 2020 eruption, a larger difference
was found, between 3 and 4±3km. The comparison of the daily mean SO₂ layer heights further demonstrates the capabilities
of this near-real-time product, with slopes between 0.8 and 1 and correlations ranging between 0.6 and 0.8. Comparisons
between the S5P+I: SO₂ LH and the CALIOP/CALIPSO ash plume height are also satisfactory at -2.5±2km, considering that



the injected SO₂ and ash plumes' locations do not always coincide over an eruption. Furthermore, the CAMS assimilation of
the S5P+I: SO₂ LH product led to much improved model output against the non-assimilated IASI layer heights, with a mean
difference of 1.5±2km compared to the original CAMS analysis, and improved the geographical spread of the Raikoke volcanic
plume following the eruptive days.

## 1    Introduction

Ten years have passed since the ash cloud from the 2010 Icelandic Eyjafjallajokull volcano caused an unprecedented disruption
to air traffic across Europe, affecting the flight schedules of approximately 10 million passengers and resulting in nearly 2
billion US dollars in lost airline revenue (Bolić and Sivčev, 2011).  This eruption led to increased awareness of the threat of
volcanic ash to air traffic in Europe, and numerous advances have taken place since then with regard to research, regulation,
and cooperation (Reichardt et al., 2017). Apart from the ash cloud, the volcanic sulphur dioxide (SO₂) plume is also hazardous
to aircraft, as it forms the corrosive sulphuric acid and can further deposit sulphates in the engines (Prata, 2009). Since the ash
particles will deposit faster than SO₂, after the first post-eruption hours, the two clouds typically separate in elevation, making
the reliable detection, dispersal and forecast of both clouds during significant explosive eruptions on a global basis equally
important (ICAO, 2012).
The disruption that the Eyjafjallajökull & Grímsvötn 2010 and 2011 eruptions had on airborne traffic has led the International
Civil Aviation Organization, ICAO, to change the previous zero tolerance policy on volcanic ash to establishing ash
concentration thresholds over Europe. Zehner et al., 2012, have translated these thresholds into specific requirements for
improved volcanic ash monitoring and forecasting services. These include the early detection of volcanic emissions and the
near real-time, NRT, global monitoring of volcanic plumes, with open access and delivery of data (Brenot et al., 2014; 2021),
but also the quantitative retrievals of volcanic ash as well as SO₂ concentration and altitude from satellite instruments, and
their validation.
While quantifying the SO₂ load emitted during explosive eruptions provides insight into volcanic processes assists in volcanic
hazard mitigation and permits the climatic impact quantification of major eruptions (Carn et al., 2016), the retrieval of the SO₂
plume injection height drives the majority of current scientific advancements in the field. Numerous eruptions have already
been used as demonstrational case studies using a variety of space-borne observations and modelling techniques to infer the
layer height such as eruptions by Mt Etna, Italy, (Boichu et al., 2015), Nabro, Erithrea, (Clarisse et al., 2014), Jebel at Tair,
Yemen (Eckhardt et al., 2008), Eyjafjallajökull and Grimsvötn, Iceland (Carboni et al., 2016), Calbuco, Chile (Pardini et al.,
2018), to name but a few.
Within the European Space Agency (ESA) Sentinel-5p+ Innovation SO₂ Layer Height project (S5P+I: SO2LH) activities have
led to the generation of a near-real-time SO₂ Layer Height product based on the Sentinel-5P/TROPOMI observations, hereafter
referred to as S5P SO₂ LH. In this work, we present the direct validation of the retrieved SO₂ layer heights for four recent





major eruptions against independent satellite information as well as its indirect verification via its assimilation into the Copernicus Atmospheric Monitoring Service, CAMS, forecast system.

## 2 S5P SO₂ Layer Height

The retrieval of the SO$_2$ layer height based on Sentinel-5P/TROPOMI measurements is performed using the already established "Full-Physics Inverse Learning Machine" algorithm (hereafter referred to as FP_ILM). The FP_ILM algorithm for the retrieval of S5P+I SO$_2$ LH is based on Hedelt et al., 2019 and is an improvement of the FP_ILM algorithm developed by Efremenko et al., 2017 for the retrieval of the SO$_2$ LH based on the Global Ozone Monitoring Experiment, GOME-2, instrument data using a Principal Component Regression (PCR) technique. In general, the FP_ILM algorithm creates a mapping between the spectral radiance and atmospheric parameters using machine learning methods. The main advantage of the FP_ILM algorithm over classical direct fitting approaches is that the time-consuming training phase involving complex Radiative Transfer (RT) modelling and Neural Network (NN) training is performed offline; the final trained inversion operator itself is robust and computationally simple and therefore extremely fast and can be applied in near-real-time (NRT) processing environments, as discussed in detail below. The FP_ILM algorithm was originally developed for the retrieval of cloud properties (Loyola et al., 2006) and has also been used for the retrieval of ozone profile shapes (Xu, et al., 2017) as well as the retrieval of surface properties accounting for bidirectional reflectance distribution function (BRDF) effects (Loyola, et al., 2020.) Recently, Fedkin et al., 2020 have applied the FP_ILM algorithm to retrieve the SO$_2$ LH based on Ozone Monitoring Instrument, OMI/Aura, observations.

The S5P SO$_2$ LH algorithm was further optimized in the framework of the ESA S5P+I: SO2LH project. The S5P+I project has been initiated to develop novel scientific and operational applications, products and retrieval methods that exploit the potential of the Sentinel-5p mission's capabilities beyond its primary objective and has been kicked-off at the end of June/beginning of July 2019. It will run until the end of 2021 and addresses seven themes related to atmospheric composition and ocean colour. The SO2LH theme is dedicated to the generation of an SO$_2$ layer height product for Sentinel-5p considering data production timeliness requirements. More details about the project can be found on the ESA S5P+I website (https://eo4society.esa.int/projects/sentinel-5p+innovation/, last access: 14.10.2021) as well as on the dedicated SO$_2$ LH project website (https://atmos.eoc.dlr.de/so2-lh/, last access: 14.10.2021), where all algorithm and product related documents are publicly available.

### 2.1 The optimised FP_ILM algorithm description

The FP_ILM SO$_2$ LH algorithm combines a Principal Component analysis (PCA) and a Neural Network (NN) approach to retrieve the SO$_2$ LH based on Sentinel-5P/TROPOMI backscattered UV Earthshine measurements in the wavelength range between 311 and 335 nm. The PCA is used to reduce the dimensionality of the high-resolution spectral measurements and to





extract the information related to the LH, whereas the NN is used to directly retrieve the LH based on the extracted principal
components (PCs) and other input parameters.
In a first step, the FP_ILM algorithm is trained using synthetic spectral UV data generated with the Linearized Discrete
Ordinate Radiative Transfer (LIDORT) model including inelastic rotational Raman scattering (RRS) implementation (Spurr
et al., 2008). About 500,000 reflectance spectra on a smart parameter grid (Loyola et al., 2016) in the wavelength range 311 -
335 nm have been generated, which are then convolved with the TROPOMI Instrument Spectral Response Function (ISRF).
This simulated dataset is split into two datasets: 90% is used for training the PCA and NN and the remaining 10% are set aside
and are used as an independent test dataset to determine the accuracy of the FP_ILM training. A PCA is then applied to the
training dataset to extract the first N=10 principle components to reduce the dimensionality of the spectral dataset. By thus
characterizing the set of simulated measurements with fewer parameters, a simpler, more stable and computationally efficient
inversion scheme can be realized.
In the second step, the PCs of each training sample along with the total ozone vertical column density ($O_3$ VCD), viewing
angles, surface pressure and albedo are used as input to train a feedforward artificial NN, with the corresponding $SO_2$ LH of
each training sample as the output layer. The NN consists of two hidden layers consisting of 40 nodes in the first and 10 nodes
in the second layer. A hyperbolic tangent layer activation function (tanh) is used and a regularization is applied to prevent the
NN from overfitting and to reduce the generalization error. Put together, the trained PCA operator and the trained NN form
the FP_ILM inversion operator, which is then applied to real spectral measurements in the operational phase.
In the operational phase, the trained PC operator is applied to TROPOMI spectral measurements which feature enhanced $SO_2$
levels, such as after a volcanic eruption, to extract the first 10 PCs and thus reduce the spectral dimension. With this information
(along with the other measured input parameters) the trained NN inverse function is then applied to retrieve the $SO_2$ LH. Note
that neither the $SO_2$ SCD nor the $SO_2$ VCD are input to the NN since they depend on the $SO_2$ LH both directly and indirectly
via the Air Mass Factor calculation and the temperature dependency of the absorption cross-section at the $SO_2$ layer altitude.
In the operational TROPOMI/S5P ground segment, Level 2 (L2) data is generated within 3 hours after sensing. Once this L2
data is available and a volcanic eruption occurs, the $SO_2$ LH algorithm is able to retrieve the corresponding layer height within
a few milliseconds per ground pixel. Even for a huge volcanic eruption with an $SO_2$ cloud spanning about 3% of the entire
orbit (i.e. about 50,000 pixels), the whole $SO_2$ LH retrieval is performed within 3 minutes. Note that the largest volcanic
eruptions detected by satellites so far (e.g., Raikoke, Kasatochi, Sarychev, Nabro) lead to typically 1-3% of ground pixels to
be processed for a limited number of orbits. The FP_ILM algorithm is several orders of magnitude faster than any of the direct
fitting approaches for UV layer height retrievals developed so far.
Closed-loop retrievals with the independent test dataset show that the $SO_2$ LH can be retrieved with an accuracy of less than 2
km for $SO_2$ VCD > 20DU (see Hedelt et al., 2019; $SO_2$ LH Algorithm Theoretical Baseline Document, ATBD, Hedelt et al.,
2021 and $SO_2$ LH Validation Report, VR, Koukouli et al., 2021). Note here that in the presence of volcanic ash, which can be
initially collocated with the $SO_2$ cloud in the young volcanic plume, the retrieved $SO_2$ LH can be underestimated by several



kilometres since the FP_ILM inversion operators were trained without taking ash absorption into account (see an extensive
discussion in SO$_2$ LH ATBD, Hedelt et al. 2021).
From the analysis presented in the SO$_2$ LH VR (Koukouli et al., 2021) it was deduced that the optimal accuracy was achieved
when filtering the reported LH values using a QA value (indicating the quality of the retrieval) greater than 0.5, a LH flag
(indicating warnings and errors during the retrieval) less than 16 and an associated SO$_2$ load greater than 20 D.U. For the
comparison against the independent datasets, the SO$_2$ LH were then gridded onto a 0.1x0.1° spatial plane at 6h intervals per
eruptive day.

## 3  Comparative datasets

Two different IASI/Metop SO$_2$ layer heights (LHs) are used for the validation of the S5P SO$_2$ LHs: the EUMETSAT ACSAF
Brescia v201510 product (Clarisse et al., 2012; 2014; Astoreca et al., 2018), here after IASI ULB/LATMOS, as well as the
University of Oxford product (Carboni et al., 2012; 2016), hereafter IASI AOPP. The two IASI approaches vary to such an
extent, as is discussed below, that we can assume that they provide two semi-independent datasets available for the validation
of the S5P SO$_2$ LHs. In addition, the CALIOP/CALIPSO space-born lidar observations of the ash plume (Winker et al., 2012;
Prata et al., 2017) will be compared to the S5P SO$_2$ LHs for the case of the Raikoke stratospheric eruption. Furthermore, the
S5P SO$_2$LH product was assimilated into a Copernicus Atmosphere Monitoring Service, CAMS, experiment (Inness et al.,
2021), and the assimilated fields were compared to the independent IASI ULB/LATMOS observations, indirectly validating
the S5P SO$_2$ LH v4.0 product.

### 3.1  IASI ULB/LATMOS SO$_2$ Layer Height dataset

The IASI/MetOp SO$_2$ ACSAF column data are fully described in Clarisse et al., 2012, where a algorithm for the sounding of
volcanic SO$_2$ plume above ~5 km altitude was presented and applied to IASI. The algorithm is able to view a wide variety of
total column ranges (from 0.5 to 5000 D.U.), exhibits a low theoretical uncertainty (3–5 %) and near real time applicability
and was thence demonstrated on the eruptions of Sarychev in Russia, Kasatochi in Alaska, Grimsvötn in Iceland, Puyehue-
Cordon Caulle in Chile and Nabro in Eritrea. Furthermore, an expansion of the algorithm to also provide SO$_2$ LHs for the
Nabro eruption using forward trajectories and CALIOP coincident measurements is described in Clarisse et al., 2014. The
IASI ULB/LATMOS dataset includes five SO$_2$ column data at assumed layer heights of 7, 10, 13, 16 and 25 km, as well as a
retrieved best estimate for the SO$_2$ LH. It is important to note that the SO$_2$ LHs provided by this algorithm are quantized every
0.5km, which renders simple scatter-type comparisons not as straightforward. This dataset is publicly available from
https://iasi.aeris-data.fr/
The observations by all Metop IASI instruments were treated as one, gridded onto a 0.1x0.1 grid at 6h intervals or each day.
The choice of the temporal field was applied since the S5P and Metop orbits differ on average by 3-4h and this temporal range





was found to be the optimal trade-off between resulting in a successful collocative dataset while also ensuring the comparisons
view the same parts of the $SO_2$ plumes. Recall also that IASI, an infrared sounder, also performs observations 12h later, during
night-time. For high enough latitudes, the time zones collapse onto another, so in the case of high latitude volcanoes, such as
Raikoke, a collocation closer in time can be achieved. For this dataset, the reported $SO_2$ LHs were restricted to altitudes less
than 25km where a successful $SO_2$ column retrieval was performed.

### 3.2 IASI AOPP $SO_2$ Layer Height dataset

The University of Oxford employs an optimal estimation scheme (Carboni et al. 2012; 2016) to estimate the $SO_2$ column
amount, the height of the $SO_2$ profile and the surface radiating temperature from IASI/MetOp-A, /MetOp-B & /MetOp-C
measurements. The Oxford retrieval has two steps. Firstly, a linear retrieval developed by Walker et al. (2011; 2012) is applied.
In the retrieval scheme a detection is considered 'positive' if the output of the linear retrieval is greater than a defined positive
threshold (0.49 effective DU, following Walker et al. 2012). The detection limits are variable-dependent on the height of the
plume and the atmospheric conditions. For a standard atmosphere (with no thermal contrast) the detection limits are estimated
to be: 17 DU for a $SO_2$ plume between 0-2 km, 3 DU between 2-4 km, and 1.3 DU between 4-6 km (Walker et al., 2011). The
detection scheme can miss part of an $SO_2$ plume under certain circumstances, such as low-altitude plumes, conditions of
negative thermal contrast (i.e. where the surface is colder than the atmosphere), and where clouds are present above the $SO_2$
plume, masking the signal from the underlying atmosphere. Secondly, an iterative retrieval is performed for the pixels that
provide positive detection results. The scheme iteratively fits the forward model (simulations) with the measurements, through
the error covariance matrix, to seek a minimum of a cost function. The forward model is based on RTTOV (Radiative Transfer
for TOVS) which is a very fast radiative transfer model for passive visible, infrared and microwave downward-viewing satellite
radiometers, spectrometers and interferometers (Saunders et al., 1999). The error covariance matrix used is the 'global error
covariance matrix' described by Carboni et al., 2012, defined to represent the effects of atmospheric variability not represented
in the forward model (FM), as well as instrument noise. A quality control is usually applied to the dataset; these are values
where the minimization routine converges within 10 iterations, the $SO_2$ amount is positive, the plume pressure is between 0
and 1100 mb and the cost function is less than 10. A comprehensive error budget for every pixel is included in the retrieval.
The IASI $SO_2$ retrieval is not affected by underlying clouds.
The IASI/AOPP dataset was also gridded onto a 0.1x0.1 grid at 6h intervals per eruptive day. An additional filter was applied
if the $SO_2$ LH ≤ 25km, the $SO_2$ LH error ≤ $SO_2$ layer height and the retrieved altitude ≠ apriori altitude at 400 mbars, which
would indicate that the retrieval reverted back to the a priori for lack of signal in the measurement.

### 3.3 CALIOP/CALIPSO Volcanic Layer Height dataset

CALIPSO (*Cloud-Aerosol and Lidar Infrared Pathfinder Observations)*, is a joint NASA/CNES (Centre National d' Études
Spatiales) satellite and part of the A-Train constellation of satellites. It is designed to study aerosols and clouds and aims to





provide profiling information at a global scale for improving our knowledge and understanding of the role of the aerosols in
the atmospheric processes. The main instrument, CALIOP (*Cloud-Aerosol Lidar with Orthogonal Polarization)*, is a dual-
wavelength (532 and 1064 nm) elastic backscatter lidar with the capability of polarization-sensitive observations at 532 nm
(Winker et al., 2010). The high-resolution profiling ability coupled with accurate depolarization measurements make
CALIPSO an indispensable tool to monitor specific aerosol species and clouds (Liu et al., 2008). The optical properties
retrieval is based on the successful cooperation of three modules whose main mission objective is to produce the CALIPSO
Level 2 data. CALIPSO is the first polarization lidar to provide global atmospheric measurements and is able to identify
volcanic eruption plumes related to the $SO_2$ Layer Height identification and retrieval (e.g. Fedkin et al., 2021; Hedelt et al.,
2019; Koukouli et al., 2014; Tournigand et al., 2020). The CALIPSO observations close to the volcanic source can be employed
in $SO_2$ LH validation studies, since ash (and/or aerosols) are initially collocated with the $SO_2$ cloud, before the gas and ash
plumes separate. Note that the footprint of CALIOP measurements is only 100m, hence the global coverage is very low and
detection of a volcanic ash plume is rare.
CALIPSO data consist of three basic types of information: (a) layer products, (b) profile products and (c) the vertical feature
mask (VFM). Layer products provide layer-integrated or layer-averaged properties of detected aerosol and cloud layers. Profile
products provide retrieved extinction and backscatter profiles within these layers. Because information on the spatial locations
of cloud and aerosol layers is of fundamental importance, the VFM was developed to provide information on cloud and aerosol
locations and types. Layer properties include layer top and base altitude, as well as physical properties of the feature such as
the Integrated Volume Depolarization Ratio, some of which are described below. Layer top and base altitudes are reported in
units of kilometres above mean sea level. Between -0.5 km and ~8.2 km, the vertical resolution of the lidar is 30-meters. From
~8.2 km to ~20.2 km, the vertical resolution of the lidar is 60-meters. Above ~20.2 km, the vertical resolution is 180-meters.
The on-board averaging scheme provides the highest resolution in the lower troposphere where the spatial variability of clouds
and aerosols is the greatest and coarser resolutions higher in the atmosphere The CALIPSO data products used in this validation
study are summarized in Table 1.

**Table 1. CALIOP/CALIPSO parameters used in this study.**

| Parameter | Version | Level | Resolution Due to Averaging | |
|---|---|---|---|---|
| | | | **Horizontal** | **Vertical (<8km)** |
| Total_Attenuated_Backscatter_532 | v.4.10 | 1 | 1/3 km | 30 m |
| Extinction_Coefficient_532 | v.3.41, v.4.20 | 2 | 5 km | 60 m |
| Aerosol Layer_Top/Base_Altitude | v.3.41, v.4.20 | 2 | 5 km | 30 m |
| Feature_Clarification_Flags | v.3.41, v.4.20 | 2 | 5 km | 60 m |




The CALIPSO version 4 (V4) product determines the locations of layers within the atmosphere, discriminates aerosols from
clouds and categorizes aerosol layers as one of eleven subtypes, seven in the troposphere and four in the stratosphere (Omar
et al., 2009; Kim et al., 2018) providing also the optical depth of each detected aerosol layer (Winker et al., 2012). The most
fundamental update in V4 is that aerosol layers are now classified as either tropospheric aerosol or of certain stratospheric
aerosol feature types. The tropospheric aerosol types include the following sub-types: clean marine, dust, polluted,
continental/smoke, clean continental, polluted dust, elevated smoke and dusty marine. Stratospheric aerosol subtypes have
been introduced for ash, sulfate/other, smoke and polar stratospheric aerosol. Note that below the tropopause, ash and sulphate
plumes are given by the tropospheric aerosol subtypes: volcanic ash is often classified as dust or polluted dust and volcanic
sulphate is often classified as elevated smoke. As a result, contiguous aerosol features crossing the tropopause will have aerosol
subtypes which switch from tropospheric to stratospheric subtypes, depending on the relationship between the attenuated
backscatter centroid altitude of the layer identified by the feature finder and the tropopause altitude. Refer to the Data Quality
Summary Document for further details (Vaughan et al., 2020).

### 3.3.1    CALIOP weighted extinction height

An important indicator for vertical profiles is the weighted extinction height, a parameter that gives in a single number an
indication of the altitude of the detected aerosol plume distribution. This parameter is considered ideal for comparisons with
aerosol layer height from passive satellite sensors (e.g. GOME-2, IASI, TROPOMI and the future Sentinel missions, since
these retrievals are very sensitive to the location of the aerosol mass maximum within the detected layers. For the validation
of the TROPOMI $SO_2$ LH, we used CALIOP level 2 version 4.10 aerosol extinction profiles at 5 km spatial resolution, retrieved
from CALIOP observations of attenuated backscatter at 532 nm (Winker et al., 2010). To facilitate quantitative comparison of
aerosol altitude, we used a mean extinction height calculated from the CALIOP extinction profile, following Koffi et al. (2012):
$$ALH_{ext} = \frac{\sum_{i=1}^{n} \beta_{ext,i} \cdot Z_i}{\sum_{i=1}^{n} \beta_{ext,i}} \qquad \text{Equation 1}$$
where $Z_i$ is the height from sea level in the $i^{th}$ lidar vertical level i (km), and $\beta_{ext,i}$ is the aerosol extinction coefficient ($km^{-1}$) at
the same level. In the CALIOP level 2 products, aerosol extinction is only retrieved for the layers in which aerosols are detected,
depending on the instrument's signal-to-noise ratio (SNR). In the case when aerosols are present over clouds, $ALH_{ext}$ will be
situated in the centre of the aerosol layer, with any undetected aerosol layers below the cloud layer not included in the
calculations due to attenuation of the signal beyond the cloud layer. According to this validation method, the CALIOP 532nm
channel observations are chosen for analysis as the conclusions from the analysis of the results do not change when the 1064
nm channel observations are used instead (Nanda et al., 2020).





# 4    Results

## 4.1    Comparisons with the IASI/Metop SO₂ Layer Heights

### 4.1.1    Raikoke, 2019

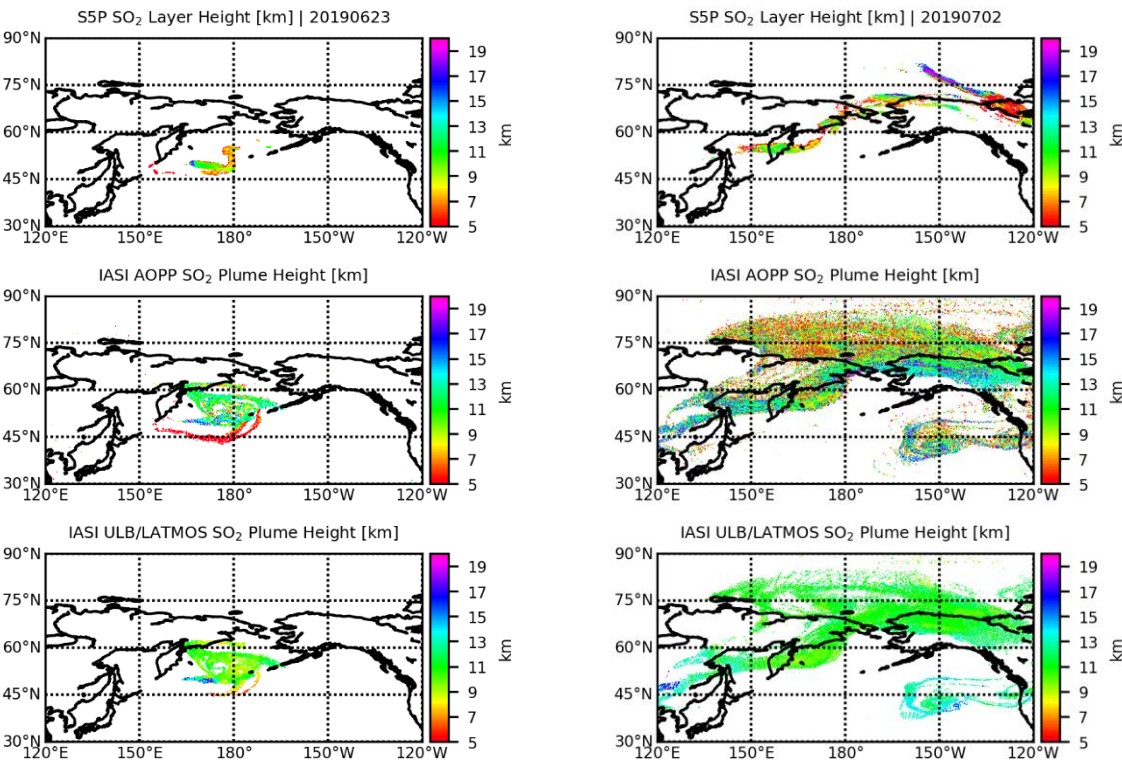

**Figure 1.** SO₂ Plume Height for two example days of the Raikoke 2019 eruptive period, the 23ʳᵈ of June on the left and the 2ⁿᵈ of July on the right. The S5P+I: SO₂ LH at the top, IASI AOPP LH in the middle and IASI ULB/LATMOS LH on the bottom panels, both ascending and descending orbits.

On June 22nd, 2019, a vast plume of ash and volcanic gases with more than 1000 DU of SO₂ was emitted during the eruption of the Raikoke volcano, Kuril Islands (McKee et al., 2021). This eruption could be detected even two months after the end of eruptive event, which rendered it an important case study for testing different satellite observations retrieval methods; the original FP_ILM methodology applied to TROPOMI observations (Hedelt et al., 2019), a probabilistic enhancement method using the Cross-track Infrared Sounder (CrIS) on the Joint Polar Satellite System (JPSS) series of satellites (Hyman and Pavolonis, 2020), a synergistic analysis of different satellite observations and dispersion modelling (Kloss et al., 2021) and the recent application of the FP_ILM algorithm to OMI/Aura observations (Fedkin et al., 2021.) This eruption was also used

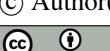
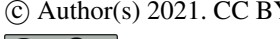



in numerical atmospheric modelling in simulating the dispersion of the Raikoke $SO_2$ cloud in the UK Met Office Numerical
Atmospheric-dispersion Modelling Environment (de Leeuw et al., 2021) and the Copernicus Atmosphere Monitoring Service
(Inness et al., 2021).
In Figure 1, two example days of the 2019 Raikoke eruption, the 23$^{rd}$ of June (left) and the 2$^{nd}$ of July (right) are shown for the
S5P $SO_2$ LH (upper), the IASI AOPP LH (middle) and the IASI ULB/LATMOS LH (bottom) observations. These
demonstrational figures do not represent collocative datasets, but rather show the spatial extent of the plumes reported by each
dataset, after filtering and gridding are performed.  Due to the restriction in $SO_2$ load necessary (> 20 D.U.) in the S5P $SO_2$
LH algorithm, the thinner parts of the plumes are not captured by the S5P observations, however its near-real-time capabilities
renders it an excellent tool for early detection in view of aviation safety. The equivalent maps for the $SO_2$ load are presented
in Figure S1, where it is shown that the extensive plumes reported by both IASI products are associated with loads of less than
~20 D.U.
The vertical distribution of the Raikoke $SO_2$ plume can be examined in the integrated $SO_2$ mass profiles presented in Figure 2.
The reported $SO_2$ load was integrated every 1km, between 0 and 20km, on the collocated gridded datasets. In these two eruptive
days, we note how the $SO_2$ mass dispersed is placed with respect to the retrieved layer height among the three datasets. Overall,
the location of the peak $SO_2$ mass is within 2km between S5P and IASI, however for the case of the IASI AOPP the amount
of ejected $SO_2$ mass is systematically lower in magnitude, even though it is well placed in height. This is most likely linked to
the quality control applied to the IASI AOPP $SO_2$ results which excludes a number of pixels within the core part of the plume,
due to the poor fit between the measured and modelled spectra.

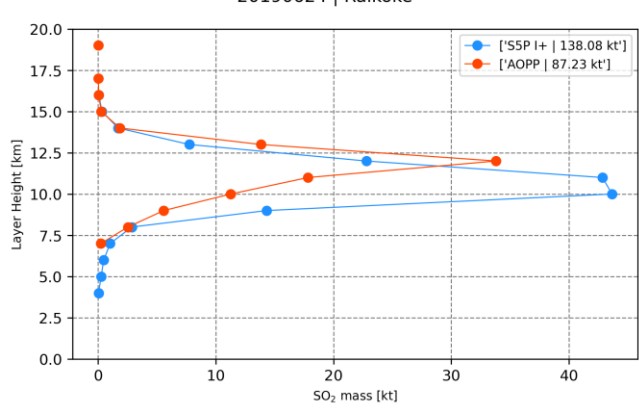
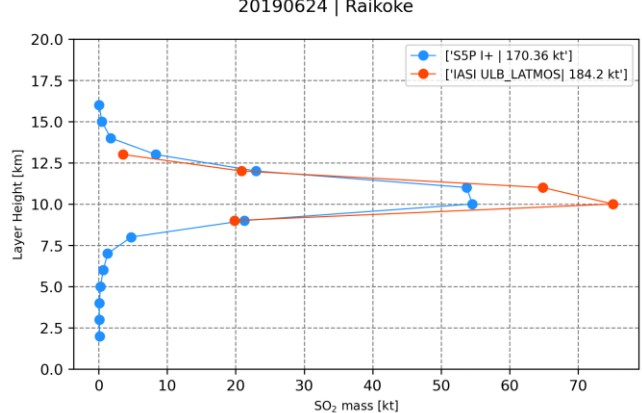



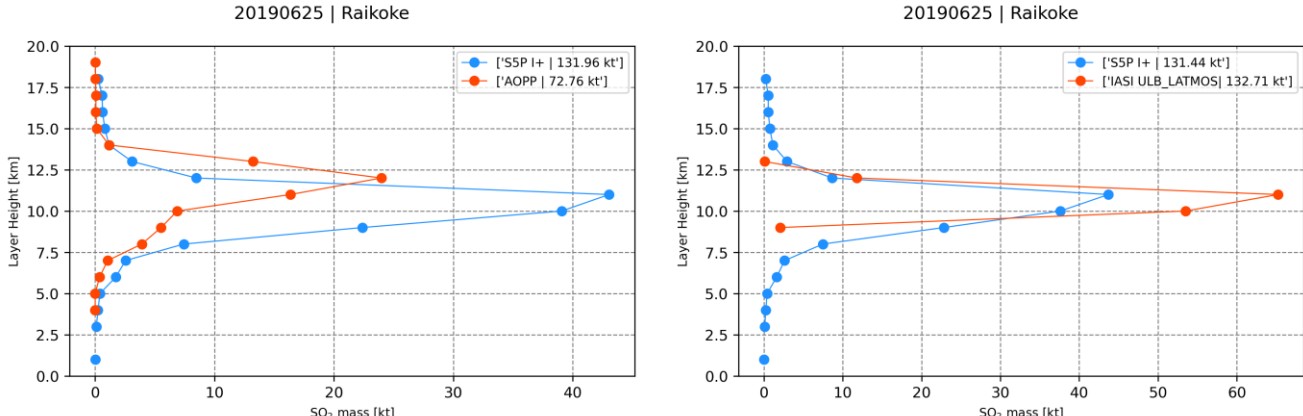

**Figure 2.** $SO_2$ integrated mass (kt) against plume altitude (km) for two example days of the Raikoke 2019 eruptive period, the 24[th] (upper row) and the 25[th] of June (lower row) for the S5P+I: $SO_2$ product in blue and the IASI AOPP in red (left column) and IASI ULB/LATMOS in red (right column). In each set, the respective collocations are shown.

Figure 3 shows the comparisons for the entire Raikoke eruptive period between the S5P $SO_2$ LH and the IASI/AOPP PH (left) and the IASI ULB/LATMOS PH (right) in histogram mode. For both comparisons, the mean S5P $SO_2$ LH is reported at 10±3km, with IASI/AOPP placing the plume at 10±1km and IASI ULB/LATMOS at 11±2.5km, resulting in an excellent mean difference between sensors of ~ ±0.5±3km.

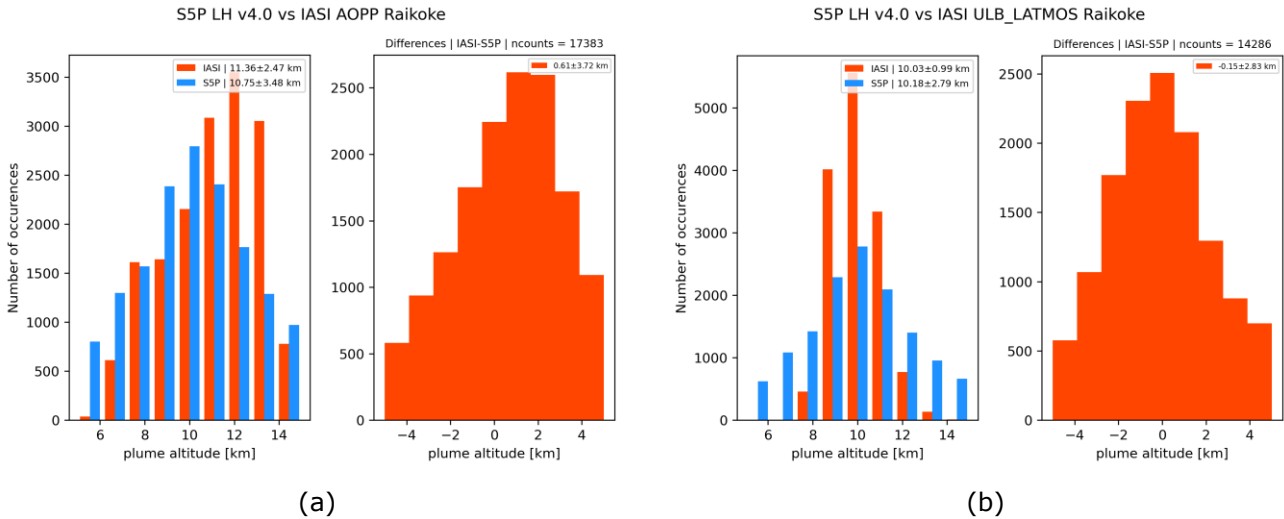

**Figure 3.** Comparisons between spatiotemporally collocated plume heights for the Raikoke, 2019, eruptive days. (a), left panel, histogram distribution for the S5P LHs (blue) and the IASI/AOPP LHs (orange) and right panel, their absolute differences. (b) as per (a) for the comparisons to the IASI ULB/LATMOS dataset.





### 4.1.2 Taal, 2020 and La Soufrière, 2021 eruptions

The Taal volcano in Batangas, Philippines erupted on the afternoon of January 12[th], 2020, 43 years after its previous eruption in 1977. Stronger explosions began around 3 pm and spewed an ash column exceeding a kilometre high. By 7:30 pm, volcanic activities intensified as continuous eruptions generated a tall 10 to 15 kilometres steam-laden tephra column (Jing et al., 2020). Perttu et al., 2020, analysed infrasound observations to the East of the volcano and estimated a plume height and duration for further ash dispersion modelling, reporting the plume at a mean height of 15km. The High Spectral Resolution Lidar of the Manila Observatory (http://www.observatory.ph/2020/01/17/taal-volcano-2020-eruption-impact-on-air-quality-part-i/, last access 13.10.2021) reported a massive ash cloud ingested and transported above the 12km altitude in the first post eruption hours, a finding further corroborated by the volcanic ash detected by the Advanced Meteorological Imager on board the GEOKOMPSAT-2A platform (Ahn et al., 2021) whose analysis also placed the ash cloud at 12km. The presence of ash hinders the detection of the $SO_2$ cloud by both UV-visible and infrared sensors and partially explains the larger spread in reported $SO_2$ layer heights by TROPOMI and IASI shown in Figure S2. A large disagreement on the location of the $SO_2$ plume is found between datasets in this case, with differences between -3 and -5km between the observations, also attributable to the ~3h difference in sensing time and its importance when studying the first few hours after a volcanic eruption (see maps in Figure S3).

On the morning of April 9th 2021, the La Soufrière volcano on the Caribbean island of Saint Vincent began erupting, spewing ash at least 7.5 km in the air, for the first time since 1979. The volcano continued to erupt over the next several days, with multiple violent explosions. Ash blanketed Saint Vincent and winds carried ash to Barbados, about 120 miles east. The Smithsonian Institute Global Volcanism Program, https://volcano.si.edu/volcano.cfm?vn=360150, last access: 13.10.2021, reported a period of explosive activity and strong pulses of ash emissions at 03:30 on the 10[th] April, whose resulting ash plumes rose to ~10-16 km altitude throughout the day. On the 12[th] of April, at 04:15, another large explosion produced an ash plume that rose to ~13 km altitude. The spread of the $SO_2$ plume sensed by TROPOMI and both IASI algorithms is shown in Figure S4, where the $SO_2$ plume reached very high altitudes, above 15km, when close in location to the volcano and decreasing in height as it progressed to the East over the sea. For both comparisons in Figure S5, the agreement of the collocative datasets is within 1km, all instruments placing the $SO_2$ plume at an average height of 14-15km.

### 4.1.3 Summary of the comparisons with the IASI/Metop observations

The overall statistics for the comparisons of the $SO_2$ plume altitude for four eruptions between 2019 and 2021 for S5P and the IASI AOPP comparisons are shown in Table 2 while those of the IASI ULB/LATMOS are given in Table 3. The collocations refer each time to those of each of the two sets. Note that for the Nisinoshima, Japan, eruptive period in July & August 2020, collocations are only available for the IASI ULB/LATMOS datasets. Overall, per eruptive period, the mean plume altitudes are similarly placed by both UV-visible and infrared instruments, with a mean difference within 1km, albeit a high standard deviation, between 2.5 and 4km.



**Table 2. Overall statistics for the comparison between S5P and IASI AOPP for the eruptive periods.**

|  | Mean S5P LH | Mean IASI AOPP LH | Mean Difference | Collocations no. |
|---|---|---|---|---|
| **Raikoke, 2019** | 10.75±3.48km | 11.36±2.47km | 0.61±3.72km | 17383 |
| **Taal, 2020** | 10.14±3.5km | 5.64±1.5km | -4.49±2.82km | 47 |
| **La Soufriere, 2021** | 13.82±2.49km | 13.47±3.41km | -0.35±3.55km | 25 |


**Table 3. Overall statistics for the comparison between S5P and IASI ULB/LATMOS for the eruptive periods.**

|  | Mean S5P LH | Mean IASI ULB/LATMOS LH | Mean Difference | Collocations no. |
|---|---|---|---|---|
| **Raikoke, 2019** | 10.18±2.79km | 10.03±0.99km | -0.15±2.83km | 14286 |
| **Taal, 2020** | 12.13±3.95km | 9.51±1.78km | -2.62±3.0km | 17 |
| **Nishinoshima, 2020** | 7.73±1.97km | 8.0±1.04km | 0.27±2.79km | 11 |
| **La Soufriere, 2021** | 14.94±3.87km | 15.7±1.16km | 0.76±3.69km | 168 |


The comparisons between S5P and IASI AOPP $SO_2$ LHs is shown, in Figure 4, left, and IASI ULB/LATMOS on the right, for
all eruptive days where the mean plume height reported for each of the 27 days of collocations is shown as a scatter plot. For
the IASI AOPP $SO_2$ LHs, left, the comparison is very promising, with a slope close to 0.9, y-intercept of 1.2km and correlation
coefficient of 0.66 for the 27 collocations days for the Raikoke, Taal and La Soufriere eruptions. The outlier point, where S5P
reports a high layer height at ~10km while IASI AOPP reports low at ~5km, belongs to the Taal comparison, discussed
previously. For ULB/LATMOS comparison, the mean $SO_2$ LHs, as expected, follow quite closely a straight line, with slope
of ~1 and y-intercept of ~0.8km, and a satisfactory correlation coefficient of 0.73. Nearly 20 days belong to the Raikoke
eruptive period, and the rest to the Taal, Nishinoshima (only for ULB/LATMOS) and La Soufriere eruptions.

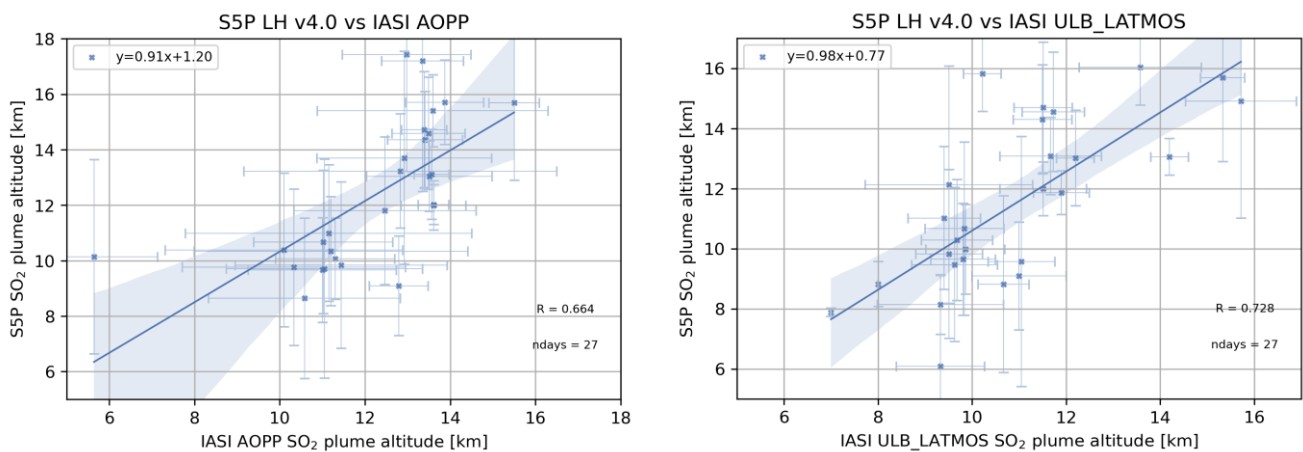

**Figure 4.** Scatter plot of the mean daily average reported $SO_2$ LHs by TROPOMI/S5P and IASI/AOPP (left) and IASI ULB/LATMOS
(right) for all available collocated eruptive days. The standard error bars represent the standard deviation of the mean.



### 4.2    Comparisons with CALIOP/CALIPSO Volcanic Ash Layer Height

#### 4.2.1    Raikoke, 2019

Within this study, the availability of overpasses of CALIPSO/CALIOP after the eruption of the Raikoke volcano on the 22nd of June was examined. Volcanic ash and sulphate aerosols are identified in CALIOP profiles based on collocated TROPOMI pixel values. The closest distances between the CALIOP footprint of the CALIPSO overpass and the locations of the TROPOMI centre pixels are selected respectively, to create collocated datasets, usually with the two orbits being within 1h to one another. To illustrate the reliability of the TROPOMI $SO_2$ LH product, we discuss in detail a selected case of collocated and concurrent TROPOMI – CALIPSO observations close to the detected $SO_2$ plume from the Raikoke eruption, on the 25rd of June 2019.

We use the 532 nm Total Attenuated Backscatter (TAB) data version 4.10 from one CALIPSO orbit in order to detect the aerosols and clouds and their heights. The TAB signal strength (Figure 5, top) is color-coded in a manner that the blue background represents molecular and weak aerosol scattering while aerosols typically  appear in the shades of red, orange and yellow. The grey scales represent the stronger cloud signals, while the weaker cloud signals, being similar in strength to the strong aerosol signals, also appear in the shades of red, orange and yellow. The TAB is sensitive to both water and ice droplets, as well as numerous types of atmospheric particles. The equivalent VFM image (Figure 5, middle) shows the aerosol type, which is retrieved according to the aerosol classification algorithm for all the detected aerosol layers. The VFM describes the vertical and horizontal distribution of both aerosols and clouds. After detection of the aerosol features, they are then classified into types and subtypes. As shown in Figure 5 (bottom), the plume scene is well captured and according to the V4 algorithm, is classified as volcanic ash/ sulphate (Kim et al., 2018). The volcanic plume of the 25th of June 2019 is marked with a dashed red circle.



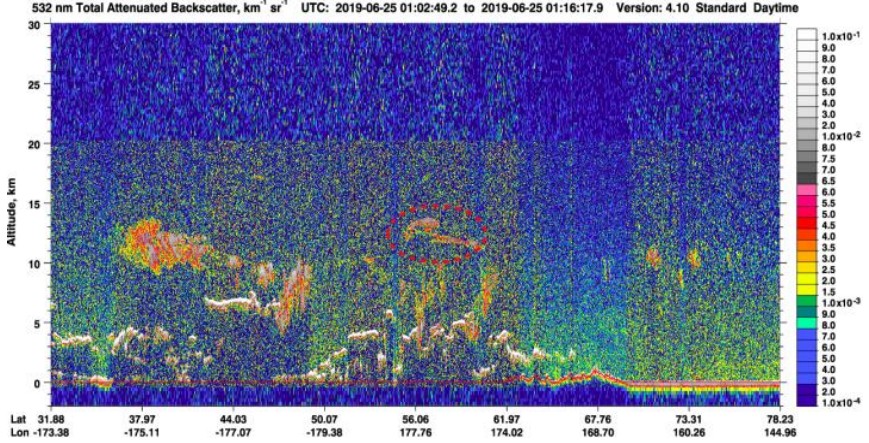

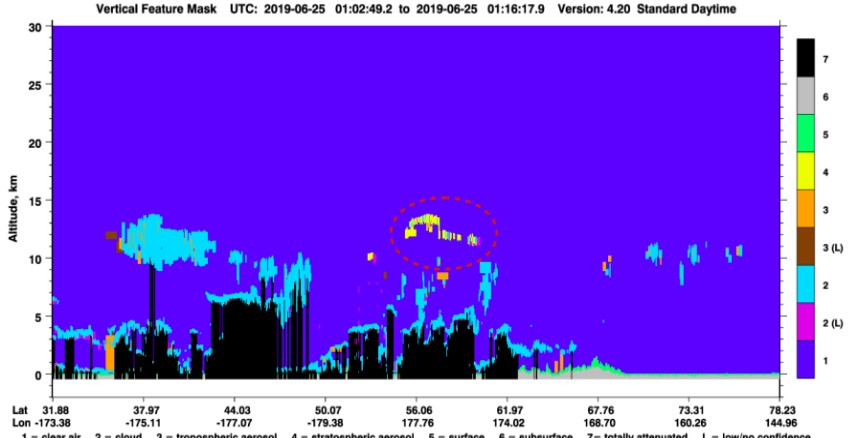

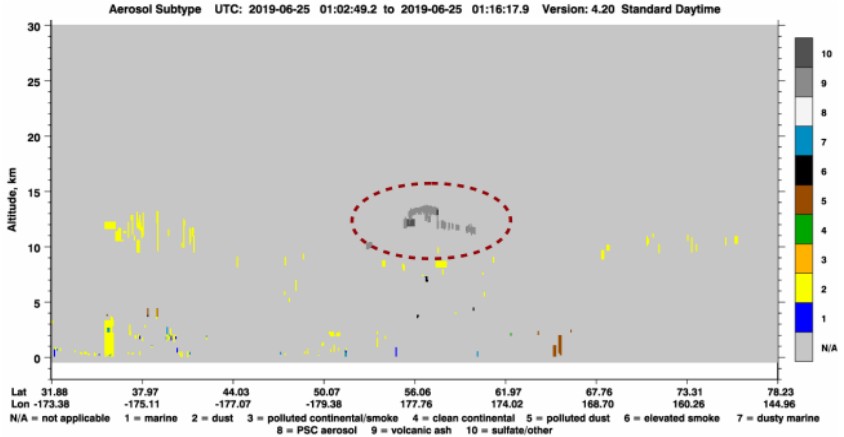





**Figure 5.** (Top) CALIOP total attenuated backscatter profile for the Raikoke eruption on the 25[th] of June 2019, (middle)
Vertical feature mask image showing the location of all layers detected and (bottom) aerosol subtype. The red dashed circles
denote the volcanic feature detected from CALIOP. (images from *https://www-calipso.larc.nasa.gov/products/*)
Figure 6 shows the TROPOMI $SO_2$ layer height pixels retrieved by the FP_ILM algorithm for $SO_2$ VCDs greater than or equal
to 20 DU, QA > 50 and LHflag < 16, overlaid with the calculated CALIPSO weighted extinction ALH pixel values (coloured
circles) which are color-coded according to the range of height values (in km). The CALIOP overpass time of this area is
between 01:00 and 01:15 UTC, and the TROPOMI overpass time is between 01:25 and 01:30 UTC, a time difference of mere
minutes. The TROPOMI plume shows several layers with $SO_2$ layer heights ranging from 5-6 km up to 14 km for this day. In
the area of the plume observed by both TROPOMI and CALIOP (54 – 58°N & 176 – 178°E), the CALIOP vertical feature
mask and aerosol subtype mask identify some volcanic ash at approximately 13 km altitude, and meteorological clouds mixed
with tropospheric aerosols (dust, polluted dust and elevated smoke) at lower altitudes. The clouds below the ash plume are
shown in blue in Figure 5, middle panel.

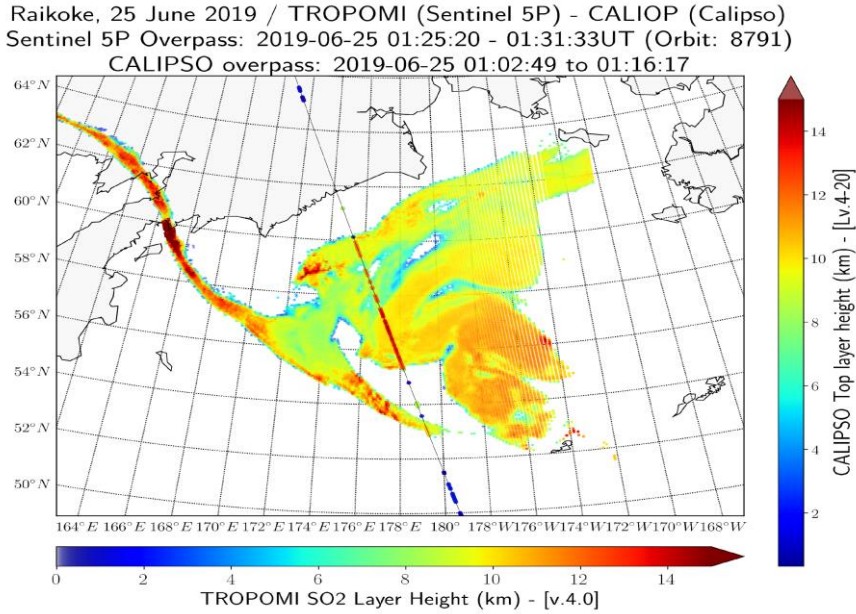

**Figure 6.** TROPOMI $SO_2$ layer height for the Raikoke volcanic eruption, measured on the 25[th] of June 2019. Only pixels with
$SO_2$ VCDs greater than or equal to 20 DU are shown. The black line indicates the CALIPSO ground track and the coloured
circles along the line indicate weighted extinction height product values (in km), for the results shown in Figure 5.

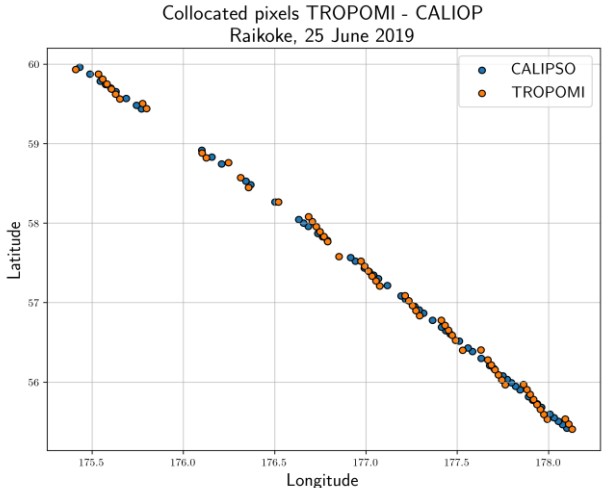
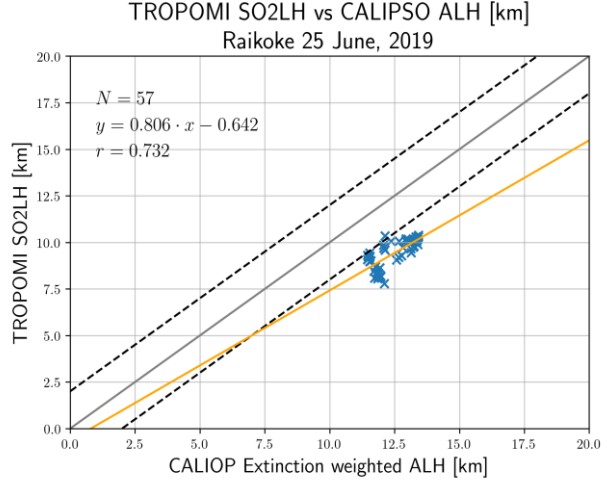

**Figure 7.** Left. The latitude/longitudes of the collocated pixels. Right. Comparison between TROPOMI SO$_2$ LH and CALIPSO weighted extinction height for the 25$^{th}$ of June 2019. The orange line is the regression line of the TROPOMI-CALIPSO observations; the grey line is the 1:1 line.

The spatiotemporal collocation between TROPOMI and CALIOP on that day is near perfect (Figure 7, left) and the spatial agreement between SO$_2$ LH and CALIOP weighted extinction altitude is satisfactory, confirming the presence of volcanic plumes. Both instruments yield high altitude values, however TROPOMI retrieves higher altitudes especially for the western part of the plume. A comparison scatterplot of collocated ash-flagged pixels is shown in Figure 7, right. The pixel-by-pixel scatter of the 57 common points shows a high correlation of 0.73, even though the SO$_2$ plume is placed approximately 2km lower that the ash plume.

Overall, seven TROPOMI (at 22/6 02:20; 23/6 00:20; 24/6 00:00; 25/6 01:30; 28/6 02:00; 29/6 02:00 and 30/6 01:30) and CALIPSO collocated overpasses (at 22/6 02:30; 23/6 01:30;24/6 00:30; 25/6 01:00; 28/6 03:00; 29/6 03:35 and 30/6 02:40) were identified. A statistical analysis has been performed using all resulting 241 collocated pixels for the 22$^{nd}$, 23rd, 24$^{th}$, 25$^{th}$, 28$^{th}$, 29$^{th}$ and 30$^{th}$ of June 2019. Figure 8 shows the distribution of TROPOMI SO$_2$ LH and CALIOP calculated weighted height differences for all days of collocation, as a scatter plot on the left and on a histogram representation on the right. The agreement is quite satisfactory with mean and median residual values around ~-2.4km and ~-3km respectively, and standard deviation of ~1.7km.

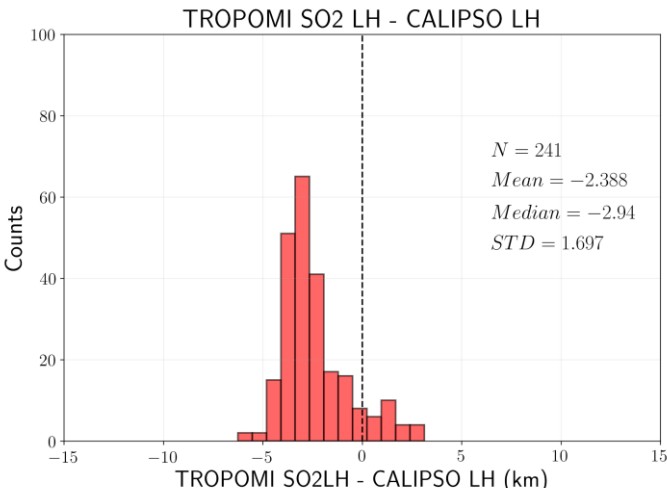
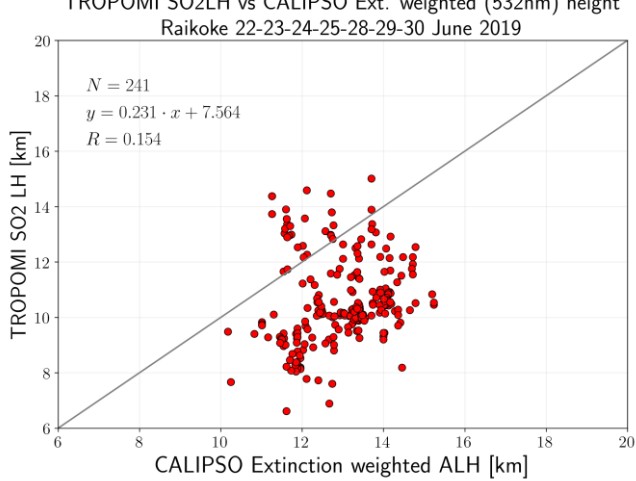

**Figure 8.** (Left) Scatter plot of the TROPOMI SO$_2$ LH and CALIPSO weighted height for all collocated pixels on the 22$^{nd}$, 23$^{rd}$, 24$^{th}$, 25$^{th}$, 28$^{th}$, 29$^{th}$ and 30$^{th}$ of June 2019, for the Raikoke eruption. (Right) Histogram distribution of the absolute differences between TROPOMI SO$_2$ LH and the corresponding CALIPSO weighted extinction height measurements, calculated for the 241 collocated points.

### 4.2.2    Sinabung, 2018, Nishinoshima, 2020 and La Soufrière, 2021 eruptions

On the 19$^{th}$ of February 2018, at 08:53 L.T., the Indonesian stratovolcano Mount Sinabung on Sumatra (2460 m summit elevation) erupted jetting a large ash plume that quickly rose to a heights of approximately 15 to 17km.. Although the eruption was spatiotemporally small an excellent overpass was found against the CALIPSO instrument (Figure S6, left). The CALIOP track crossed the main part of the volcanic cloud, across the north-to-south axis.Its overpass time is between 07:08 and 07:22 UTC, a mere 45 min after the the TROPOMI overpass time, between 06:24 and 06:26 UTC. The CALIPSO observations showed both the ash cloud, as a layer around 5 km, as well as two vertical ash clouds extending from the volcano up to ~10 km altitude. As shown in **Figure 9**, where the S5P SO$_2$ LH retrievals are shown in the red dots, the presence of clouds appear along the CALIPSO path indicated by the stronger attenuated backscatter than the aerosol layer.

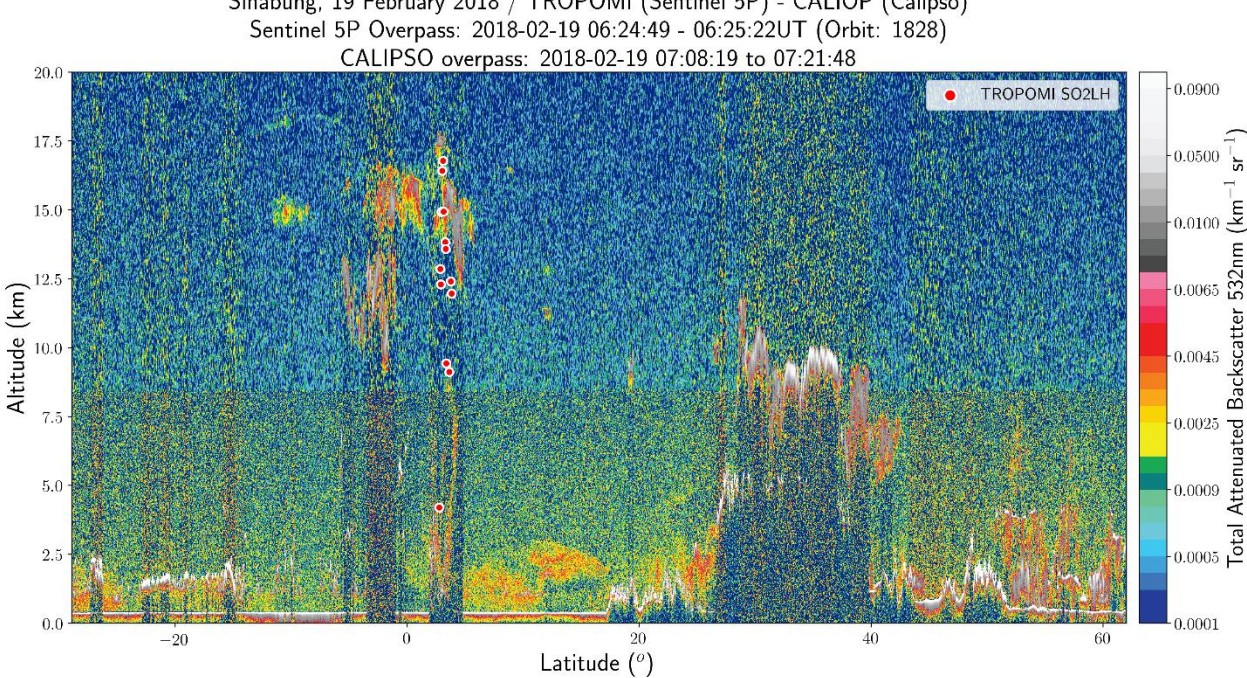

**Figure 9.** Sinabung, 19th of February 2018, 07:15 UTC. The colours show the CALIOP/CALIPSO total attenuated backscatter at 532nm and the white-red dots show the TROPOMI SO$_2$ LH.

This case of mixing between ash and clouds over a volcanic eruption renders the retrieval of the ash plume altitude by the lidar algorithm very difficult, since it cannot separate clouds from aerosols, especially when the aerosol amount is low. The CALIPSO feature mask (not shown here) hardly identifies any of the Sinabung backscatter signals as aerosol. The main plume, at ~15km is flagged a cloud feature, while below this feature everything is masked as "totally attenuated", which is not expected to be the case. Most probably liquid water or ice particles are contaminating the volcanic ash signal, as already discussed in Hedelt et al., 2019. Even though the maximum TROPOMI SO$_2$ LH agrees with the maximum backscatter height between 2-3° latitude, a large spread of TROPOMI SO$_2$ LHs are also reported. As discussed also in the work of de Laat et al., 2020, the presence of either a nearly-transparent or a bright cloud may result in the TROPOMI algorithm reporting heights far lower than both the ash and the cloud plumes. For the cases of Nishinoshima 2020 and La Soufrière 2021 eruptions, both provided a satisfactory collocation to the CALIOP orbital path without the difficulties found in the case of Sinabung, 2018, enabling a meaningful comparison to be made. For Nishinoshima, spatial collocations for the 1[st] of August 2020 are shown in Figure S7 (left), while the scatterplot of height values is shown in the right. The geographical collocations between TROPOMI and CALIOP are not optimal, however the agreement between SO$_2$ LH and CALIOP weighted extinction altitude is satisfactory, and tends to confirm the presence of volcanic plumes. The CALIPSO observations confirm the presence of volcanic clouds





around 5 km, while S5P reports slightly higher loads, at ~7.5km. For the case of La Soufrière, spatial collocations for the 11st
of April 2021 are shown in Figure S8 (left), where the scatterplot of collocations is shown in the right column and the scatter
plots in the right column. In this case, both CALIPSO and TROPOMI collocated pixels confirms the presence of a volcanic
cloud up to and around ~20km.

### 4.2.3    Summary of the comparisons with the CALIPSO/CALIOP observations

The combination of CALIOP and TROPOMI data measurements has permitted the identification of volcanic aerosol layers
produced by three individual volcanic eruptions. A summary plot of the comparisons between S5P $SO_2$ and CALIPSO ash
LHs is presented as a scatter plot in Figure 10, showing the mean ash and $SO_2$ plume height reported for each of the 9 days of
collocations. The comparison is very promising, with a slope close to 0.95, y-intercept of ~1 km and correlation coefficient of
0.86 for the 9 collocations days for the Raikoke, Nishinoshima and La Soufrière eruptions. The majority of cases, 7 days,
belong nearly to the Raikoke eruptive period, and the remainder 2 days to Nishinoshima and La Soufrière eruptions,
respectively. From Table 4 it is worth noting that the standard deviation of the mean heights reported by both instruments are
low, typically much less than 1km. This can most likely be attributed to the tight spatiotemporal collocation criteria that were
possible for these comparisons.

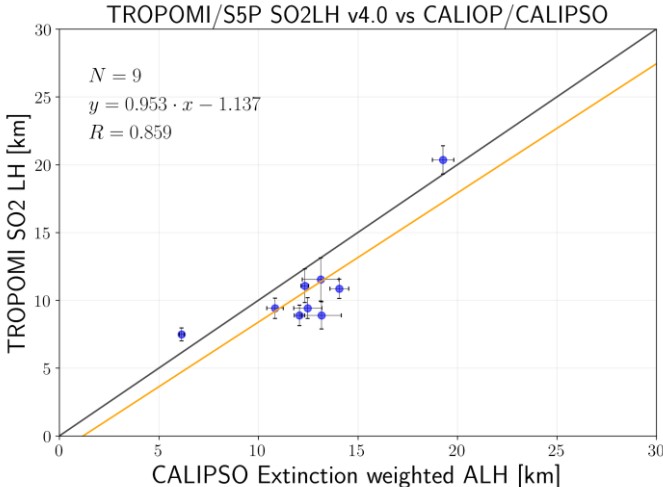

**Figure 10.** Scatter plot of the mean daily average reported $SO_2$ LHs by TROPOMI/S5P and CALIOP/CALIPSO for the seven

days of the Raikoke eruption and one each for Nishinoshima and La Soufrière eruptions studied.

**Table 4. Statistics for the comparison between S5P and CALIPSO for the eruptive days studied.**

| Eruptive day | Mean CALIPSO LH [km] | Mean S5P Height [km] | Mean Difference [km] | Collocations. |
|---|---|---|---|---|



| 22 June 2019 | 10.84±0.4 | 9.40±0.75 | -1.43±0.56 | 8 |
|---|---|---|---|---|
| 23 June 2019 | 12.06±0.28 | 8.88±0.76 | -3.17+0.98 | 13 |
| 24 June 2019 | 12.33±0.2 | 11.07±1.24 | -1.26±1.40 | 22 |
| 25 June 2019 | 12.47±0.1 | 9.41±0.76 | -3.05±0.54 | 57 |
| 28 June 2019 | 13.12±0.92 | 11.53±1.6 | -1.59±2.13 | 87 |
| 29 June 2019 | 14.06±0.47 | 10.84±0.7 | -3.21±0.99 | 46 |
| 30 June 2019 | 13.16±1 | 8.88±1 | -4.28±0.56 | 8 |
| 01 August 2020 | 6.14±0.12 | 7.48±0.48 | 1.34±0.46 | 8 |
| 11 April 2021 | 19.28±0.54 | 20.35±1.04 | 1.06±1.44 | 12 |

Generally, we note that features identified as volcanic ash by the CALIOP aerosol subtype mask are captured by the TROPOMI
algorithm, but the surrounding clouds often affect the retrieval. The comparison of the TROPOMI $SO_2$ LH product within this
project shows promising capability in detecting plumes of volcanic origin, with some limitations related to existing or
subsequent creation of clouds. Furthermore, although ash and $SO_2$ plumes are often collocated especially at the first hours after
eruptions, this is not always the case, making direct comparisons challenging.

**4.3 Application of the S5P SO2 LH in NRT data assimilation modelling**

The Copernicus Atmosphere Monitoring Service (CAMS), operated by the European Centre for Medium-Range Weather
Forecasts (ECMWF) on behalf of the European Commission, provides daily $SO_2$ analyses and 5-day forecasts of volcanic $SO_2$
in NRT by assimilating total column $SO_2$ retrievals from TROPOMI and GOME-2 (Inness et al., 2021). As the operational
NRT TROPOMI and GOME-2 retrievals do not provide any information about the height of the volcanic plumes, the $SO_2$
increments are placed in the mid-troposphere, around 550 hPa (~5 km) in the current operational CAMS configuration.





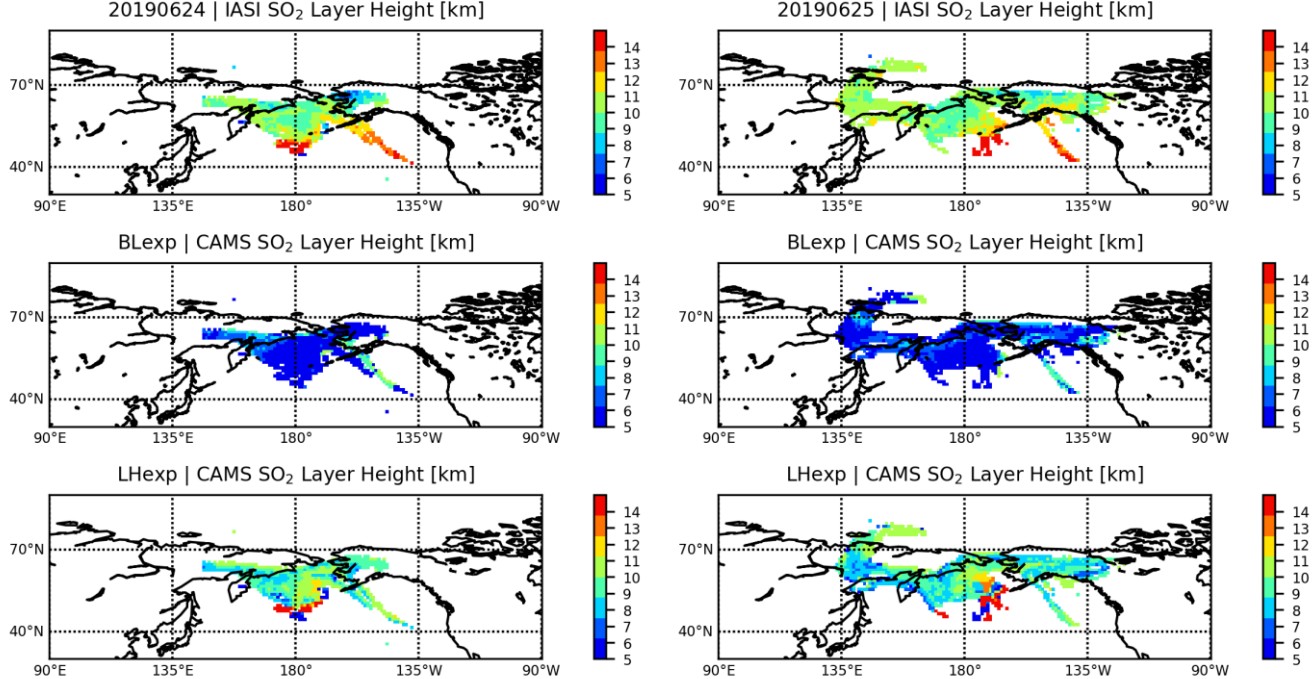

**Figure 11.** Raikoke eruptive day of the 24th (left) and 25th (right) of June 2019. Top. The IASI ULB/LATMOS SO$_2$ layer height in km. Middle. The CAMS BLexp SO$_2$ layer height. Bottom. The CAMS LHexp SO$_2$ layer height.

In the recent paper by Inness et al., 2021, the procedure used to assimilate near-real time TROPOMI/S5P and GOME2/Metop SO$_2$ loads in the operational CAMS NRT data assimilation system was presented, alongside the simultaneous ingestion of the S5P SO2LH discussed in this work. The assimilation of the S5P SO$_2$LH data was based on a previous version of the dataset, v3.1, and not the final one, v4.0, presented in this work. The assimilation was tested for the 2019 Raikoke eruption and was contrasted to the operational CAMS forecasts obtained when assimilating only the TROPOMI SO$_2$ load. Two example days are shown in this paper. In Figure 11, upper, the IASI ULB/LATMOS SO$_2$ layer height, gridded onto the CAMS 1x1° spatial resolution and 3h temporal resolution, is shown for the 24th (left column) and the 25th (right column) of June 2019, 5 days after the initial Raikoke eruption. In the middle panel, the operational CAMS SO$_2$ layer height (called BLexp) is presented which is deduced from placing the SO$_2$ increment in the mid-troposphere, around 550 hPa, clearly in the wrong altitude for the Raikoke eruption which injected a huge amount of SO$_2$ above the tropopause, well into the stratosphere. In Figure 11, lower panel, it can be seen that a vast improvement to the CAMS forecast is achieved for both days when the S5P SO$_2$ LH data are used (called LHexp) as the structure of the Raikoke SO$_2$ plume is much improved and compares well with the independent IASI SO$_2$ layer heights shown in the upper panel. For the entire eruptive period of Raikoke between June 22nd and June 29th, the CAMS forecast which assimilates the S5P SO$_2$ LH data improves the bias in the forecast height between CAMS and IASI to ~ -1.5±2.5km, compared to a mean bias of ~ -5±2km for the operational system. We can hence conclude that by assimilating the S5P SO$_2$LH data, the vertical location of the Raikoke SO$_2$ plume in the CAMS system is improved, leading to better





subsequent forecasts (Inness et al., 2021) and making the S5P SO$_2$ LH product suitable for NRT assimilation and forecasts of
a possible strong future volcanic eruption.
**5      Conclusions**
The European Space Agency Sentinel-5p+ Innovation TROPOMI/S5P SO$_2$ layer height product has been verified against
IASI/Metop SO$_2$ layer heights for the eruptive periods of the Raikoke volcano, 22 June to 30 July 2019, the Taal volcano, 13
January 2020, the Nishinoshima eruptive period during July & August 2020 and the La Soufrière eruptive days of April 10$^{th}$
to 11$^{th}$, 2021. Two different algorithms that provide plume altitude from the IASI instruments were examined, the official
EUMETSAT ACSAF algorithm, ULB/LATMOS, and the University of Oxford, AOPP, algorithm. Furthermore, collocations
against ash layer height observations by the space-born CALIOP/CALIPSO lidar system were identified and assessed.
The main findings in the comparisons of the SO$_2$ volcanic plumes, described in detail above, are:
▪    For the Raikoke eruptive days: the difference between S5P and IASI/AOPP SO$_2$ LH datasets is 0.61±3.72km, with

IASI/AOPP SO$_2$ LH reporting a mean height of ~11.5±2.5km and S5P reporting ~10.5±3.5km, in excellent agreement.

Between S5P and IASI ULB/LATMOS SO$_2$ LHs a similar mean difference of ~0±3km is found with both sensors

reporting on average LHs at ~10km.

▪    For the Taal eruptive day: the SO$_2$ LHs reported differ substantially with IASI/AOPP reporting heights ~5.5±1.5km while

S5P reports higher columns, at ~10±3.5km. IASI ULB/LATMOS also reports lower heights, at 9.5±2km while and S5P

places the plume at ~12±4km with a mean difference of~2.5±3km.

▪    For the Nishinoshima eruptive days: both sensors place the plume at the same altitude, with IASI ULB/LATMOS at

~8±1km and S5P ~8±2km and mean difference of~0±3km.

▪    For the La Soufrière eruptive days: both sensors report high plume altitudes, at ~15km, with both IASI/AOPP and

ULB/LATMOS standard deviation at ~1km and the S5P standard deviation at ~4km, and overall mean difference of

~1±3.5km.

▪    Scatter plot comparisons of the daily mean volcanic SO$_2$ plumes  reveal common SO$_2$ LHs patterns for the two sensors,

with substantial correlations ~0.66 (0.72), slope ~0.9 (0.98), y-intercept of 1.2km (0.8km) for the IASI/AOPP and the

IASI ULB/LATMOS respectively. The standard deviation of the mean is relatively high, on average ~3km, however the

mean heights are well within the 2km accuracy requirement on the S5P SO$_2$ layer height product.

With respect to the comparisons between the S5P SO$_2$ LH and the CALIOP/CALIPSO volcanic ash layer height, we report
that:
▪    241 excellently spatiotemporally collocated points between CALIOP and TROPOMI were identified for seven Raikoke

eruptive days. CALIOP reported a range of mean heights between ~11 and 14km, while TROPOMI had a far narrower

range between ~9 and 11.5km. Overall, the mean difference in heights was found to be -2.4±1.7 km (-3.0km median) for

the seven eruptive Raikoke days.



▪   The comparisons for the Nishinoshima and La Soufrière eruptions showed good agreement with plumes reported (low) at
~7.km (~19.5km) respectively for the two eruptions, and a height difference between S5P and CALIPSO being within
~1.0km.
▪   The mean daily height comparative plot of the comparisons between S5P SO$_2$ LHs and CALIOP/CALIPSO weighted
ALH, as expected, follow quite closely a straight line, with slope of 0.95 and y-intercept of ~1.0km and excellent
correlation coefficient at 0.86.
Finally, the CAMS assimilation of the NRT S5P SO$_2$ LH led to much improved model fields against the non-assimilated IASI
plume heights for the Raikoke eruptive period, with a mean difference of 1.5±2km against the independent IASI/Metop
observations, and improved the geographical spread of the Raikoke volcanic plume following the main eruptive day.

**Data availability.** The near-real-time S5P SO$_2$ LH products are operationally generated by DLR in the framework of the
Innovative Products for Analyses of Atmospheric Composition, INPULS, project, and are available upon request from Pascal
Hedelt (Pascal.Hedelt@dlr.de). The IASI/MetOp ULB/LATMOS open source SO$_2$ layer height dataset is publicly available
from https://iasi.aeris-data.fr/so2_iasi_a_arch/ (last access: 20.07.2021). The IASI/MetOp AOPP SO$_2$ products are available
on request from Isabelle Taylor (isabelle.taylor@physics.ox.ac.uk). The CALIPSO data were obtained from the online archive
of the NASA Langley Research Center Atmospheric Science Data Center (ASDC,
https://asdc.larc.nasa.gov/project/CALIPSO). The Copernicus Atmosphere Monitoring Service is operated by the European
Centre for Medium-Range Weather Forecasts on behalf of the European Commission as part of the Copernicus program
(http://copernicus.eu) and CAMS data are freely available from atmosphere.copernicus.eu/data. The SO$_2$ analysis experiments
used in this paper are available from https://apps.ecmwf.int/research-experiments/expver/ with the DOIs: 10.21957/cygt-xf49
(BLexp), 10.21957/qfam-7474 (LHexp).

**Acknowledgments** This work is performed in the framework of ESA's Sentinel-5p+ Innovation: SO$_2$ Layer Height project
(S5P+I: SO$_2$ LH), https://eo4society.esa.int/projects/sentinel-5p-innovation-so2-layer-height-project/. The comparative results
presented in this work have been produced using the Aristotle University of Thessaloniki High Performance Computing
Infrastructure and Resources. M.E.K. would like to acknowledge the support provided by the IT Center of the Aristotle
University of Thessaloniki throughout the progress of this research work, as well as the Atmospheric Toolbox®. I.A.T. and
R.G.G. would like to acknowledge EUMETSAT for providing the IASI spectra and ECMWF and CEDA for the meteorological
profiles used in the IASI retrievals. I.A.T. and R.G.G. further acknowledge support from the NERC Centre for Observation
and Modelling of Earthquakes, Volcanoes, and Tectonics (COMET). We thank the DLR Innovative Products for Analyses of
Atmospheric Composition, INPULS, project, for continuously providing the S5P SO$_2$ LH products in near-real-time.





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
