# Peer review of "Volcanic SO2 Layer Height by TROPOMI/S5P; evaluation against"

_Atmospheric Chemistry and Physics, 2021_

## Referee Comment (RC2)

Review of manuscript "Volcanic SO$_2$ Layer Height by TROPOMI/S5P; validation against IASI/MetOp and CALIOP/CALIPSO observations." by Koukouli et al.

**General comments:**

This paper compares volcanic SO$_2$ Layer Height (LH) retrievals from TROPOMI/S5P with the retrievals from the IASI/MetOp and CALIOP/CALIPSO instruments for several recent volcanic eruptions. Based on the analysis presented, the authors find that the average S5P SO$_2$ LH product compare well with the LH estimates from the IASI and CALIOP/CALIPSO instruments. Furthermore, the authors find that assimilating the S5P SO$_2$ LH product into the CAMS model improves the forecast of the 2019 Raikoke volcanic plume.

Overall, the paper provides a useful evaluation of the SP5 SO$_2$ LH product and the topic of this paper will be of interest to the readers of this journal. The results show that these new S5P plume height estimates are of equal quality as other similar products and is a useful addition to the existing range of volcanic plume height estimation techniques/products available in the literature. The paper is well structured, and the quality of the figures and tables is good.

However, I do have some concerns regarding the discussion of the results. In general, I miss an in-depth discussion of the implications of the results and the potential additional new information we can obtain from considering this new product. The paper presents the results of the comparison of S5P with the other satellite products, but not much is presented in terms of discussion of their findings and the differences found between the different retrievals. (e.g. What are the main limitations of the S5P LH product? What are the main advantages of using the S5P LH product with respect to the IASI SO$_2$ LH estimates? Why is the distribution in figure 3b for TROPOMI over a much wider range compared to the IASI estimates?)

This study also mainly focusses on the mean LH values and how they compare between the satellite products. However, for most volcanic eruptions, the emissions take place over a large range of altitudes (as is also evident by the large standard deviations reported in the manuscript, e.g. tables 2 & 3). The comparison of the distribution of the plume altitudes is therefore potentially equally relevant. Therefore, I think the paper would benefit from a more detailed discussion of the differences between the LH distributions (e.g. as shown in figure 3) and the corresponding statistics.

Furthermore, I found a lot of small inconsistencies between the values reported in the text and the tables. The level of significance for the reported values of the same quantities are not consistent (e.g. average SO$_2$ LH in tables 2 and 3, versus values in L.274-277). The reported values should be made consistent throughout the manuscript.

Finally, section 4.3 shows very similar results as presented by Inness et al. 2021. The difference is the use of a later version of the TROPOMI SO2 LH product (see L.445: "The assimilation of the S5P SO2LH data was based on a previous version of the dataset, v3.1, and not the final one, v4.0 presented in this work"). However, no information is given what the differences are between the two versions. The general conclusions presented here are

very similar as presented by Inness et al. 2021. I think a more detailed discussion of the differences between the two studies should be included, as otherwise this section provides very limited new information to the scientific community.

I would recommend major revisions before the paper can be considered for publication.

**Specific comments:**

Title: 'validation' I am not sure this is the right term. As there are a lot of uncertainties also in the IASI/MetOP and CALIOP/CALIPSO retrieval algorithms, we can't be sure they represent the true values either. Instead, 'evaluation' or 'comparison with' might be better terms to use.

L.28: 'satisfactory' How would you define if a comparison is satisfactory? It depends on the application for which you want to use the $SO_2$ LH product and is case specific.

L.31: I think there is a comma missing after '1.5±2km'.

L.42: The used reference (ICAO, 2012) doesn't mention $SO_2$ clouds and is mainly focussed on the risks posed by ash clouds. I think a different reference should be used here, for example: https://www.icao.int/airnavigation/METP/MOGVA Reference Documents/IAVW Roadmap.pdf (page 12)

L.59: 'validation' -> 'evaluation' or 'comparison'. See comment about the title.

L.261: 'Figure S1, … associated with loads of less than ~20 DU.' This is not clear from figure S1, as the colour scale range is starting at 20 DU for all panels. The only difference figure S1 shows is that the IASI estimates have a larger region where the $SO_2$ load is >20 DU. Why is the extent of the 'dense' plume in the IASI retrievals so much bigger than what is retrieved by TROPOMI?

L.267: 'well placed in height'. Is this correct? Figure 2 shows that the IASI AOPP product for the integrated $SO_2$ mass peaks 1-2 km higher than the estimates from TROPOMI and IASI ULB/LATMOS. As we are near the tropopause height, this change in peak altitude can mean the difference between most of the plume reaching the stratosphere or not. What could be the cause this difference?

L.268: If I understand correctly, when all pixels are excluded in IASI AOPP for a single grid box, the grid box is excluded from the presented comparison. Therefore, as each of the considered gridded data points is a collection of multiple pixels, could the exclusion of several pixels within a grid box explain the observed differences? Assuming that an average is calculated for each grid box, the difference seen would indicate that only very high concentration pixels are excluded by the IASI AOPP quality control. Is this the case, and if so, why would the IASI product have this bias?

L.275: 'the mean S5P SO2 LH is reported at 10±3 km'. This is not consistent with the values presented in figure 3 and table 2. According to the legend in figure 3a and table 2, the estimate is 11±3 km (using the correct rounding).

L.276: 'IASI AOPP placing the plume at 10±1 km'. How does this follow from Figure 3? I think the values for the two IASI products in the text are swapped, as it also does not correspond to the values reported in tables 2 and 3.

L.313: What are the values 2.5 and 4 km based on? In tables 2 & 3 neither of these values are present, so I am not sure I understand how these values are calculated.

L.324: The results presented in this work are heavily biased towards 1 eruption (Raikoke). How does this impact the statistics presented? If we exclude the other eruptions, what would the correlation coefficient be?

Fig.4: What are the uncertainty ranges of the slope and intercept calculated for the best linear fit? Is this represented by the blue shading in the figure? Some more explanation is needed in the caption on the blue shading and uncertainty ranges should be reported in the manuscript.

Fig.5: Please refer to figure 6 in the caption for the path of the CALIPSO satellite, as it made it easier for me to interpret figure 5.

Fig.6: Are the 2 colour scales different? I found it very confusing to have two very similar colour bars. If they are the same, I think it is better to use the same colour bar for both retrievals and have a double colour bar title instead.

Fig.7: In the right panel, what is the uncertainty in the 'best' slope calculated?

Fig. 8: The correlation between individual TROPOMI and CALIPSO pixels seems to be low when considering all the points in figure 8b. However, when considering the daily average values in figure 10, the comparison is much better. Is this because the points in fig 8b are clustered by day, therefore giving a better correlation for each of the individual overpasses? Might be useful to use 7 different colours in figure 8b to indicate the different days.

L.382: Related to the previous point, is there an impact of the aging plume on the results found? For example, can we expect the differences between CALIPSO and TROPOMI to be larger for older plumes due to the different dispersion of ash and $SO_2$?

L.446: What are the main differences between version 3.1 and 4.0? Do you expect there to be a big difference in skill between the two versions? Comparing the reported CAMS forecast bias of -1.5±2.5 km (L.475) in this manuscript with the value reported by Inness et al 2021 (0.4±2.2 km), it seems that the latest version 4.0 is less accurate. I think some discussion of this fact and potential reasons/implications should be included.

L.470: Are these LH values correct? Based on the results in tables 2 and 3, the values should be 11.4±2.5 km for IASI AOPP and 10.8±3.5 for S5P. Also, I am not sure I understand where

the 10.5 km comes from, as I can't find it anywhere in the results section (I think it should be 10.8 km based on table 2). Please check that all the values are correct.

L.471: Why are the results for the IASI ULB/LATMOS SO$_2$ LHs presented at a smaller accuracy in the conclusion section compared to table 3 (i.e. 0±3 km instead of -0.2±2.8 km)? Please make sure all the values in the text are consistent with the values presented in the tables/figures.

L.473-475: Some of the reported values for the Taal eruption are inconsistent with the values presented in table 2 and 3.

L.477: Different accuracy of the values than what is presented in table 3.

L.478: 'both sensors report high plume altitudes, at ~15km with both IASI/AOPP and ULB/LATMOS standard deviation at ~1km'. This is not consistent with the results section. Based on table 2, the LH for IASI AOPP is 13.5 km with a standard deviation of 3.4 km. Also, the S5P standard deviation is different in the two tables (2.5 km and 3.9 km) compared to the 4 km reported here.

L.491: "(low)" what does this word refer to? I think you mean that both showed low altitude plumes, but please add some additional text to clarify.

L.495: "quite closely". This is not a scientific term and should be avoided.

**Technical corrections/suggestions:**

L.25:    3km -> 3 km There are several other values in the abstract and the rest of the text where a space is missing between the value and the unit. I have tried to highlight most of them here, but please check carefully throughout the manuscript.

L.50: "volcanic processes assists in" missing comma

L.120: 20DU -> 20 DU. This is the first mention of DU, so I think it should be spelled out here.

L.127: D.U. -> DU. Throughout the manuscript both 'DU' and 'D.U.' are used. Please check and only use one consistent abbreviation.

L.157: 25km -> 25 km

L.194: 100m -> 100 m

L.225: Missing bracket after TROPOMI

L.242: '*including* both ascending'

L.264: 1km -> 1 km & 20km -> 20 km

L.274-275: PH -> LH ?

L.284: high -> thickness?

L.287: 15km -> 15 km

L.293: location -> altitude?

L.306: 1km -> 1 km & 14-15km -> 14-15 km

L.312: 1km -> 1 km

L313: 4km -> 4 km

L.373: 2km -> 2 km

L.374: that -> than

L.388: omit 'a' & 17km.. -> 17 km.

L.390: axis.Its -> axis. Its

L.393: Why is Figure 9 in bold font?

L.421: omit 'nearly'

L.423: 1km -> 1 km

Figure 11: Please expand the caption by explaining the BLexp (no assimilation) and LHexp (assimilation of TROPOMI data) terms.

L.469-498: a lot of places where a space between the value and unit is missing.

L.492: 7.km -> 7 km

L.482: 0.72 -> 0.73.  Based on L.324 I think this should be 0.73

---

## Author Comment (AC1)

Review of manuscript "Volcanic SO$_2$ Layer Height by TROPOMI/S5P; validation against IASI/MetOp and CALIOP/CALIPSO observations." by Koukouli et al.

We thank the reviewer for her/his positive comments on our work and all the suggestions which improve our work. Please find our replies insert in red.

**General comments:**

However, I do have some concerns regarding the discussion of the results. In general, I miss an in-depth discussion of the implications of the results and the potential additional new information we can obtain from considering this new product. The paper presents the results of the comparison of S5P with the other satellite products, but not much is presented in terms of discussion of their findings and the differences found between the different retrievals. (e.g. What are the main limitations of the S5P LH product? What are the main advantages of using the S5P LH product with respect to the IASI SO$_2$ LH estimates?
Why is the distribution in figure 3b for TROPOMI over a much wider range compared to the IASI estimates?)

The S5P LH product has the advantage that the LH is retrieved in the UV wavelength range, which is sensitive to other atmospheric levels than the IR based LH retrieval based on IASI data, hence different parts of the volcanic cloud are sensed. It allows for the direct determination of the proper SO$_2$ VCD of the volcanic SO$_2$ cloud, which was lacking in the past due to missing LH information. Although the IASI LH gives a first estimate of the height of the volcanic cloud, this information cannot be used in S5P SO$_2$ retrievals due to the difference in overpass time and pixel resolution. Currently, the main limitation of the S5P LH product is that it can only be applied to modest to high volcanic eruptions, with SO$_2$ VCD > 15-20 DU, hence weak volcanic eruptions cannot be considered. Furthermore the presence of ash has a strong impact on the retrieved LH, especially in the fresh plume, which however applies to all SO$_2$ LH retrievals based on satellite data.
A comment based on the above has been added in the discussion of Figure 1.

This study also mainly focusses on the mean LH values and how they compare between the satellite products. However, for most volcanic eruptions, the emissions take place over a large range of altitudes (as is also evident by the large standard deviations reported in the manuscript, e.g. tables 2 & 3). The comparison of the distribution of the plume altitudes is therefore potentially equally relevant. Therefore, I think the paper would benefit from a more detailed discussion of the differences between the LH distributions (e.g. as shown in figure 3) and the corresponding statistics.

It is indeed inevitable that the validation is biased towards Raikoke, since it was the strongest and most long-lasting eruption during the time period of our study. In one of the following replies to comment we provide statistics showing the effect the other eruptions have on the overall findings of this work.

Furthermore, I found a lot of small inconsistencies between the values reported in the text and the tables. The level of significance for the reported values of the same quantities are not consistent (e.g. average $SO_2$ LH in tables 2 and 3, versus values in L.274-277). The reported values should be made consistent throughout the manuscript.

You are correct that it is a simple case of rounding. We took the liberty to round to the nearest digit throughout the text in the hope that the take-home message would be clearer. In the tables we kept the precision of our calculations. Since this however causes confusion, we will keep the same numerals in the text, throughout. We also noticed some misplaced statistics in these lines, i.e. the ones applying to the IASI AOPP collocations were reported for IASI ULB/LATMOS, and vice versa, which were corrected. Thank you for pointing this out!

Finally, section 4.3 shows very similar results as presented by Inness et al. 2021. The difference is the use of a later version of the TROPOMI SO2 LH product (see L.445: "The assimilation of the S5P SO2LH data was based on a previous version of the dataset, v3.1, and not the final one, v4.0 presented in this work"). However, no information is given what the differences are between the two versions. The general conclusions presented here are very similar as presented by Inness et al. 2021. I think a more detailed discussion of the differences between the two studies should be included, as otherwise this section provides very limited new information to the scientific community.

The main difference between v3.1 and v4.0 was the significant increase of training samples, which was done after an internal analysis has showed that only with more than about 300,000 samples the training error converged to a minimum. Furthermore the number of nodes in the first hidden layer of the NN was slightly lower in v3.1 (32 nodes vs 40 nodes in v3.1 and v4.0, respectively). The final settings were chosen after a extensive hyperparameter optimization process.
Following similar suggestions from the second reviewer, we have re-written the entire subsection with a different focal point.

**Specific comments:**

Title: 'validation' I am not sure this is the right term. As there are a lot of uncertainties also in the IASI/MetOP and CALIOP/CALIPSO retrieval algorithms, we can't be sure they represent the true values either. Instead, 'evaluation' or 'comparison with' might be better terms to use.

A valid point. We have altered the title to:

Volcanic SO2 Layer Height by TROPOMI/S5P; evaluation against IASI/MetOp and CALIOP/CALIPSO observations.

L.28: 'satisfactory' How would you define if a comparison is satisfactory? It depends on the application for which you want to use the SO$_2$ LH product and is case specific.

You are right and we have updated the text in a more appropriate manner.

L.31: I think there is a comma missing after '1.5±2km'.

Agreed, included.

L.42: The used reference (ICAO, 2012) doesn't mention SO$_2$ clouds and is mainly focussed on the risks posed by ash clouds. I think a different reference should be used here, for example: https://www.icao.int/airnavigation/METP/MOGVA Reference Documents/IAVW Roadmap.pdf (page 12)

Thank you for this updated reference, included.

L.59: 'validation'  ->  'evaluation' or 'comparison'. See comment about the title.

We reworded to evaluation as suggested.

L.261: 'Figure S1, … associated with loads of less than ~20 DU.' This is not clear from figure S1, as the colour scale range is starting at 20 DU for all panels. The only difference figure S1 shows is that the IASI estimates have a larger region where the SO$_2$ load is >20 DU. Why is the extent of the 'dense' plume in the IASI retrievals so much bigger than what is retrieved by TROPOMI?

In Figure S1, and in all the similar map-type figures presented in the main paper as well, all values below the lowest colour level [20 D.U. for the SO$_2$ load and 5km for the SO$_2$ LH] are depicted with the colour of the lowest level. All SO$_2$ loads below 20 D.U. hence appear in the beige colour of the lowest chosen level. Both IASI retrievals provide an estimate for the SO$_2$ LH even at SO$_2$ loads below 20 D.U., which is not the case for S5P.

L.267: 'well placed in height'. Is this correct? Figure 2 shows that the IASI AOPP product for the integrated SO$_2$ mass peaks 1-2 km higher than the estimates from TROPOMI and IASI ULB/LATMOS. As we are near the tropopause height, this change in peak altitude can mean the difference between most of the plume reaching the stratosphere or not. What could be the cause this difference?

From our experience in analysing the different satellite-born LH estimates, we have reach the conclusion that a difference of 1-2 km between UV and IR sensors is acceptable. Considering the fact that S5P and IASI have completely different retrieval approaches and completely different wavelength range making them sensitive to different atmospheric layers, such differences can be explained – and have been explained - in literature. Furthermore, the inherent difficulties reported for the IASI AOPP algorithm in sensing the thickest parts of the

SO$_2$ plume, due to super-saturation effects, further explains why the IASI ULB/LATMOS plumes show their highest load at the same altitude as S5P.

L.268: If I understand correctly, when all pixels are excluded in IASI AOPP for a single grid box, the grid box is excluded from the presented comparison. Therefore, as each of the considered gridded data points is a collection of multiple pixels, could the exclusion of several pixels within a grid box explain the observed differences? Assuming that an average is calculated for each grid box, the difference seen would indicate that only very high concentration pixels are excluded by the IASI AOPP quality control. Is this the case, and if so, why would the IASI product have this bias?

Indeed, the IASI AOPP algorithm quality control rejects pixels within the core part of the plume, due to the poor fit between the measured and modelled spectra. The SO$_2$ spectral lines chosen by the IASI AOPP algorithm get saturated by the large SO$_2$ amounts and the retrieval fails to pass the quality control. This is a known fact to the IASI AOPP algorithm scientists and a different algorithm set-up to amend this issue is currently work-in-progress.

L.275: 'the mean S5P SO2 LH is reported at 10±3 km'. This is not consistent with the values presented in figure 3 and table 2. According to the legend in figure 3a and table 2, the estimate is 11±3 km (using the correct rounding).

Thank for your spotting this, the entire paragraph was updated to include the more appropriate, and correct, statistical numbers.

L.276: 'IASI AOPP placing the plume at 10±1 km'. How does this follow from Figure 3? I think the values for the two IASI products in the text are swapped, as it also does not correspond to the values reported in tables 2 and 3.

Thank for your spotting this, the entire paragraph was updated to include the more appropriate, and correct, statistical numbers.

L.313: What are the values 2.5 and 4 km based on? In tables 2 & 3 neither of these values are present, so I am not sure I understand how these values are calculated.

The appropriate mean values are now included in the paragraph.

L.324: The results presented in this work are heavily biased towards 1 eruption (Raikoke). How does this impact the statistics presented? If we exclude the other eruptions, what would the correlation coefficient be?

It is indeed inevitable that the validation is biased towards Raikoke, since it was the strongest and most long-lasting eruption during the time period of our study. Removing the two other days of eruptions from both comparisons of Figure 4 the statistics do not alter significantly. In parenthesis, I provide the statistics of Figure 4. For IASI AOPP, the slope is 0.90 [0.91], yinterscept of 1.40 [0.90] and correlation coefficient of 0.631 [0/66. For IASI ULB LATMOS, slope of 1.10 [0.98], y-interscept of -0.45 [0.77] and correlation coefficient of 0.663 [0.72].

Fig.4: What are the uncertainty ranges of the slope and intercept calculated for the best linear fit? Is this represented by the blue shading in the figure? Some more explanation is needed in the caption on the blue shading and uncertainty ranges should be reported in the manuscript.

Thank you for pointing this out. The light blue shaded areas represent the 95% confidence intervals of the fit. The information has been added in the figure caption as well as in the text describing it. We have further included the error estimated on the slope and y-intercept in the text.

Fig.5: Please refer to figure 6 in the caption for the path of the CALIPSO satellite, as it made it easier for me to interpret figure 5.

Figure caption updated as requested.

Fig.6: Are the 2 colour scales different? I found it very confusing to have two very similar colour bars. If they are the same, I think it is better to use the same colour bar for both retrievals and have a double colour bar title instead.

Thanks for the comment. The "stripes" in the TROPOMI SO2LH map in Figure 6 are due to a simple visualitaion of the TROPOMI pixels via Python. Each color grid point represents the center of TROPOMI pixel so there are "white" areas left between pixels. It is not related to any gridding process.

Fig.7: In the right panel, what is the uncertainty in the 'best' slope calculated?

The uncertainty ranges of the slope and intercept calculated for the best linear fit (y=mx+b):
- The slope uncertainty: $m_{best} \pm \Delta m = 0.8 \pm 0.10$
- The y-intercept uncertainty: $b_{best} \pm \Delta b = -0.6 \pm 1.2$

Fig. 8: The correlation between individual TROPOMI and CALIPSO pixels seems to be low when considering all the points in figure 8b. However, when considering the daily average values in figure 10, the comparison is much better. Is this because the points in fig 8b are clustered by day, therefore giving a better correlation for each of the individual overpasses? Might be useful to use 7 different colours in figure 8b to indicate the different days.

The reviewer is right, when clustering - by calculating the daily mean - the correlation is rather impressive. There are 7 days for Raikoke and 1 each for Nishinoshima and La Soufriere. Figure 8b was updated following the reviewer's suggestion, colouring each point according to the day of. Collocations hence showing the spread of the clustering for each day.

L.382: Related to the previous point, is there an impact of the aging plume on the results found? For example, can we expect the differences between CALIPSO and TROPOMI to be larger for older plumes due to the different dispersion of ash and SO₂?

Based on our results we cannot really argue that there is a "clear" influence of aging. However, taking into account previous studies related to the Raikoke eruption, we can summarize the following main points:

- The comparison results TROPOMI-CALIPSO suggests that aerosol dynamic process is crucial for the height differences. Our results reveal that the detected aerosol layers altitudes increased slightly the next days after the eruption day. Both the aging process and the aerosol radiation interaction can influence the vertical distribution of aerosols and therefore determine at which altitude the particles are transported.

- The behaviour of altitude range differences could be also explained by the results of Muser et al., 2020; De Leeuw a et al., 2020 and Osborne et al., 2021. These studies underline that for coarse mode ash the aging process is the determining factor of whether the volcanic plumes rises and sinks. As volcanic aerosols are often composed of a complex mixture of both ash and sulfate, which changes with time, the strict classification becomes more challenging.

- As volcanic aerosol layers evolve and disperse into the atmosphere their optical and microphysical properties are expected to change in time. Thus, the classification of volcanic cloud based upon their optical properties since those properties evolve with time depending on the presence of ash and sulfate which can also misclassified. CALIPSO observations of several volcanic plumes during the last years composition can vary significantly depending on the initial injection of volcanic ash and SO₂ further oxidized into sulfate. It is too simple to assume that volcanic plumes are made entirely of sulfate, even several days after the eruption.

L.446: What are the main differences between version 3.1 and 4.0? Do you expect there to be a big difference in skill between the two versions? Comparing the reported CAMS forecast bias of -1.5±2.5 km (L.475) in this manuscript with the value reported by Inness et al 2021 (0.4±2.2 km), it seems that the latest version 4.0 is less accurate. I think some discussion of this fact and potential reasons/implications should be included.

Please see our reply above to your previous comment on this topic. We can further note that during the development of the SO₂ LH algorithm it was found that, although v3.1 was slightly more accurate for some volcanic eruption events, for other events it performed extremely

poorly. In contrast, v4.0 performed well over all volcanic eruptions analysed in the time frame o this work.

L.470: Are these LH values correct? Based on the results in tables 2 and 3, the values should be 11.4±2.5 km for IASI AOPP and 10.8±3.5 for S5P. Also, I am not sure I understand where the 10.5 km comes from, as I can't find it anywhere in the results section (I think it should be 10.8 km based on table 2). Please check that all the values are correct.

It was a simple case of rounding, for reasons discussed above. The conclusions were updated to include the full accuracy statistics shown in the relevant sections of the paper.

L.471: Why are the results for the IASI ULB/LATMOS $SO_2$ LHs presented at a smaller accuracy in the conclusion section compared to table 3 (i.e. 0±3 km instead of -0.2±2.8 km)? Please make sure all the values in the text are consistent with the values presented in the tables/figures.

As above.

L.473-475: Some of the reported values for the Taal eruption are inconsistent with the values presented in table 2 and 3.

As above, with a typographical mistake as well, thank you for spotting it.

L.477: Different accuracy of the values than what is presented in table 3.

It was a simple case of rounding, for reasons discussed in the beginning of these replies. The conclusions were updated to include the full accuracy statistics shown in the relevant sections of the paper.

L.478: 'both sensors report high plume altitudes, at ~15km with both IASI/AOPP and ULB/LATMOS standard deviation at ~1km'. This is not consistent with the results section. Based on table 2, the LH for IASI AOPP is 13.5 km with a standard deviation of 3.4 km. Also, the S5P standard deviation is different in the two tables (2.5 km and 3.9 km) compared to the 4 km reported here.

As above, with a typographical mistake as well, thank you for spotting it.

L.491: "(low)" what does this word refer to? I think you mean that both showed low altitude plumes, but please add some additional text to clarify.

This is a simply typographical error from a previous version of the manuscript. Thanks for noticing.

L.495: "quite closely". This is not a scientific term and should be avoided.

Indeed, thanks for pointing this out.

**Technical corrections/suggestions:**

All the following suggestions were taken into consideration in the updated text.

L.25:    3km -> 3 km There are several other values in the abstract and the rest of the text where a space is missing between the value and the unit. I have tried to highlight most of them here, but please check carefully throughout the manuscript.

L.50: "volcanic processes assists in" missing comma

L.120: 20DU -> 20 DU. This is the first mention of DU, so I think it should be spelled out here.

L.127: D.U. -> DU. Throughout the manuscript both 'DU' and 'D.U.' are used. Please check and only use one consistent abbreviation.

L.157: 25km -> 25 km

L.194: 100m -> 100 m

L.225: Missing bracket after TROPOMI

L.242: '*including* both ascending'

L.264: 1km -> 1 km & 20km -> 20 km

L.274-275: PH -> LH ?

L.284: high -> thickness?

L.287: 15km -> 15 km

L.293: location -> altitude?

L.306: 1km -> 1 km & 14-15km -> 14-15 km

L.312: 1km -> 1 km

L313: 4km -> 4 km

L.373: 2km -> 2 km

L.374: that -> than

L.388: omit 'a' & 17km.. -> 17 km.

L.390: axis.Its -> axis. Its

L.393: Why is Figure 9 in bold font?

L.421: omit 'nearly'

L.423: 1km -> 1 km

Figure 11: Please expand the caption by explaining the BLexp (no assimilation) and LHexp (assimilation of TROPOMI data) terms.

L.469-498: a lot of places where a space between the value and unit is missing.

L.492: 7.km -> 7 km

L.482: 0.72 -> 0.73.  Based on L.324 I think this should be 0.73

---

## Author Comment (AC2)

We thank the reviewer for her/his positive comments on our work and all the suggestions which improve our work. Please find our replies insert in red.

Title: The IASI retrievals are also uncertain and quite different between the two products. Also the CALIPSO aerosol heights may or may not have the same vertical distribution as $SO_2$. Given the uncertainties in these independent datasets, one may argue that the study here is more of a "comparison" rather than a "validation".

Following a similar suggestion from the other reviewer, we have changed the title to: Volcanic $SO_2$ Layer Height by TROPOMI/S5P; evaluation against IASI/MetOp and CALIOP/CALIPSO observations.

Lines 151-154: With the different overpass times between IASI and TROPOMI, why not use trajectory model to match measurements between different sensors?

This is a very good idea. We are currently working on implementing a trajectory/dispersion model in order to track and forecast the volcanic plume. This could be also used to correct for different overpass times. We are however still in the process of developing this and cannot simply apply this in the given paper. This is nevertheless foreseen in the near future.

Figure 2: Are the integrated profiles based on the same grid cells (i.e., for grid cells that have both valid TROPOMI and IASI height retrievals)? Or is the mass difference between IASI/AOPP and TROPOMI due to different pixels being integrated? What could be the reason for different $SO_2$ mass estimates between TROPOMI and IASI/AOPP? Please clarify.

The IASI AOPP algorithm quality control rejects pixels within the core part of the plume, due to the poor fit between the measured and modelled spectra. The $SO_2$ spectral lines chosen by the IASI AOPP algorithm get saturated by the large $SO_2$ amounts and the retrieval fails to pass the quality control. This is a known fact to the IASI AOPP algorithm scientists and a different algorithm set-up to amend this issue is currently work-in-progress. As a result, when all pixels are excluded in IASI AOPP for a single grid box, the grid box is excluded from the presented comparison and, due to the particularities of the algorithm discussed above, very high concentration pixels are excluded by the IASI AOPP quality control. This in turn lowers the IASI AOPP $SO_2$ mass estimate, which is not the case for the IASI ULB/LATMOS dataset.

Figures 2 and 3: The distribution of TROPOMI retrievals is more spread-out - do we know why?

The main reasons for a broader distribution of TROPOMI retrievals wrt IASI is quite likely the different wavelength range sensing the plume as well as the use of a completely different retrieval approach, i.e. NN vs optimal estimation. In the NN L2 regularization is applied such that it generalizes better, especially since simulated reflectance spectra are used for the training of the NN. This has some impact on the spread of the LH retrievals. Secondly the IASI LH retrievals are sensitive to a different altitude range in the IR than that of the UV wavelength range used in the S5P LH retrievals, which in turn has also influence on the LH distribution.

Lines 324-325: The comparison sample is dominated by Raikoke - can the authors elaborate how this affects the comparison (for example, correlation coefficient)?

It is indeed inevitable that the validation is biased towards Raikoke, since it was the strongest and most long-lasting eruption during the time period of our study. By removing the two other days of eruptions from both comparisons of Figure 4, the statistics do not alter significantly. For IASI AOPP, the slope is 0.90 [0.91], y-intercept of 1.40 [0.90] and correlation coefficient of 0.631 [0.66]. For IASI ULB LATMOS, slope of 1.10 [0.98], y-intercept of -0.45 [0.77] and correlation coefficient of 0.663 [0.72]. In parenthesis I provide the statistics of Figure 4.

Figure 6: Please specify the thresholds used to filter out CALIPSO weighted extinction height data (clouds?)

To avoid cloud contamination of aerosol retrievals, cloud signatures must be identified and removed. Prior to analysis, advanced QA procedures are performed on the L2_05kmAPro product to remove highly uncertain aerosol extinction data values. This QA scheme is similar to that employed in Campbell et al. (2012) and Winker et al. (2013) and involves several parameters included in the L2_05kmAPro product:

- (1) Extinction_QC_532 (r) is equal to 0, 1, 2, 16 or 18,
- (2) $-20 \geq CAD\_Score(r) \geq -100$,
- (3) Extinction_Coefficient_Uncertainty_532(r) $\leq 10$ km$^{-1}$
- (4) Extinction_Coefficient_532($\geq 0$ and $\leq 1.25$ km$^{-1}$),

Further details of each QA parameter are documented in the CALIPSO Data Users Guide (http://www-calipso.larc.nasa.gov/resources/calipso_users_guide/).

Also Figure 6: Are stripes in TROPOMI SO$_2$ heights due to retrievals or gridding?

Thanks for the comment. The "stripes" in the TROPOMI SO2LH map in Figure 6 are due to a simple visualization of the TROPOMI pixels via Python. Each color grid point represents the center of TROPOMI pixel so there are "white" areas left between pixels. It is not related to any gridding process.

Figure 7: Can the authors use different colors for the data points in the right panel based on SO$_2$amount?

We have updated Figure 7 according to the reviewer's suggestion. The new figure is color-coded according to the corresponding TROPOMI SO2 VCDs while range between 21 and ~120 D.U. for that day.

Figure 8: Can the authors comment on the low correlation between CALIPSO and TROPOMI? Is the correlation coefficient a function of time since the main eruption? Based on Figure 6 and Figure 8, can we draw the conclusion that individual TROPOMI retrievals are not so well-correlated with CALIPSO measurements?

What is immediately apparent from our analysis is that the CALIOP ALHext is higher than TROPOMI SO2LH almost consistently for most of the cases As volcanic aerosol layers evolve and disperse into the atmosphere, their optical and microphysical properties are expected to change with time. We see a systematic increase in the average daily heights in CALIOP measurements, which is not the case with the TROPOMI observations (Table 4). It should also be noted that the number of collocations vary significantly day-by-day. We cannot however argue that there is a "clear" dependence over time for the correlation between the compared datasets.

The obvious reason is, as discussed in this work but also other studies that compare CALIPSO to Uv/Vis instruments volcanic observations, the $SO_2$ and ash/aerosol plumes are not necessarily collocated, as gas and ash can separate in volcanic ash, especially in the days following the eruption. After they separate, it is up to the prevailing winds in the region to either separate them further, or bring them at the same altitude once more. Ageing of the ash particles also plays a significant role, further complicating the issue.

Another reason could be due to CALIOP possibly underestimating the aerosol layer thickness due to strong attenuation of the lidar signal at the top of the layer (Rajapakshe et al., 2017), whereas the TROPOMI SO2LH product does not suffer from such attenuation. It is well known that the CALIOP based layer detection often misses the lowest boundary of a thick aerosol layer, thereby biasing the bottom of the aerosol layer high. We have to note here that the CALIOP is able to measure only ash and aerosol absorption profiles.

Section 4.2.2 focuses on Sinabung but the discussion (lines 397-405) appears to indicate that the eruption is not an ideal case for validation/comparison using CALIPSO?

From our extensive analysis of this volcanic eruption during the S5P+I: $SO_2$ LH project, we have concluded that this case is not an "ideal" case for a direct comparison of the TROPOMI-CALIPSO datasets. We selected to present this case study for Sinabung in the paper to demonstrate a situation where is characterized by complexity. This case of mixing between ash and clouds over a volcanic eruption renders the retrieval of the ash plume altitude by the lidar algorithm very difficult, since it cannot separate clouds from aerosols, especially when the aerosol amount is low. Although the eruption was spatiotemporally small an excellent overpass was found against the CALIPSO instrument. Performing the validation of the Sinabung eruption (12 January 2018) we used Total attenuated backscatter (TAB) (and show in the manuscript) to provide a qualitative analysis for the $SO_2$ LH retrievals. We summarize the main points arising from this case study:

- We expect that, in case of sufficiently dense ash, the cloud height data products provide accurate volcanic ash cloud heights. On the other hand, In case of semi-transparent volcanic ash clouds, where the cloud top height retrievals become sensitive to other reflective surrounding surfaces (water/ice clouds) the detection of accurate volcanic ash cloud heights is limited.

- Fresh volcanic layers are typically rich in water vapor (volcanic clouds also contain high concentrations of water). Due to this fact, the classification in the CALIPSO vertical feature mask sometimes fails to pick up the volcanic ash or sulfate aerosol because of

competing clouds. TROPOMI may underestimate actual ash heights in case of semi-transparent volcanic ash clouds, especially in the presence of high concentrations of water vapor and for very high-altitude volcanic ash clouds (Hedelt et al., 2019, 2021).

Additionally, we provide an example below, showing the CALIOP VFM (left) and cloud phase (right) images corresponding to the Sinabung eruption. Types of clouds are flagged in the released VFM as cirrus - ice clouds. The "detected feature" is marked with a dashed red circle.

[Figure]

Minor comments:

Line 26: What does "3 and 4±3km" mean?

It refers to 3±3 and 4±3 km; this was altered accordingly.

Line 27: Correlation coefficients?

Of course, you are right. Updated in the text.

Lines 50-52: The sentence is too long and difficult to follow. There are several other places where shorter sentences may help the readers.

Thank you, we have split this sentence into two.

Line 59: What do you mean by "direct validation"?

Direct is a validation/verification that compares two products directly, on a one-to-one basis. The word is added to show the difference to the validation/verification performed via the CAMS assimilation experiments where the S5P SO2 LH is assimilated, and the resulting LH is compared to the IASI ULB/LATMOS LH. I.e. in an indirect way.

Line 79: "has been kicked-off" should be "was kicked off".

Agreed, changed.

Line 98: " By thus" should be "By"?

Agreed, changed.

Line 179: These are the conditions under which a retrieval would be considered valid for comparison? Please clarify.

Thank you, it was indeed a confusing point. We have rephrased the entire two final paragraphs.

Line 189: what are the "three modules"? Please specify or remove the statement, otherwise it could be confusing.

The note on the "three modules" was rephrased to "three major algorithm steps" referring to the CALIPSO algorithm processing steps, in the updated manuscript. Briefly:

The level 2 processing involves three major steps. First, cloud and aerosol layers are identified by a set of algorithms, applied to the 532-nm attenuated backscatter profiles. After this, a set of scene classification algorithms (SCA) classifies these layers by type. Using data from the CALIOP channels, layers are identified as clouds or aerosols and the aerosol type and then the cloud ice–water phase are determined. At the end, profiles of particle backscatter and extinction coefficients are retrieved by the extinction retrieval algorithm performing retrievals within the layer boundaries identified before. Finally, classification procedures then proceed, layer-by-layer.

In the figure below we illustrate an example (not added in the revised manuscript) for 25 June 2019 for Raikoke. An uplifted ash layer can be seen using the CALIOP level 2 products. The (a) panel shows vertically resolved 532 nm attenuated backscatter [km-1sr-1], (b) panel show corresponding depolarization ratio, (c) the vertical feature mask and finally the (d) display the Aerosol subtype of the scene. Depolarization measurements indicate that the aerosol in the layer was predominantly spherical and therefore comprised mostly of sulfate.

[Figure]

(c) N/A = not applicable   1 = marine   2 = dust   3 = polluted continental/smoke   4 = clean continental   5 = polluted dust   6 = elevated smoke   7 = dusty marine
8 = PSC aerosol   9 = volcanic ash   10 = sulfate/other

(d) 1 = clear air   2 = cloud   3 = tropospheric aerosol   4 = stratospheric aerosol   5 = surface   6 = subsurface   7= totally attenuated   L = low/no confidence

Line 284: "a kilometre high" refers to the height or thickness of the ash plume?

The thickness, indeed. We have rephrased.

Line 370 (also lines 380-382): "satisfactory" - means the difference is within the expected uncertainty range of TROPOMI retrievals?

We consider that, especially in the comparisons with the ash plume observations by CALIPSO, the main source of uncertainty is the fact that the two plumes do not always coincide, as the separate – typically- early on after eruption. Hence, an overall difference of 2km can be considered spectacular keeping in mind of course that individual days' comparisons vary. In the new, colour-coded per eruptive day, scatter plot in Figure 8, one can note that for most of the days the comparisons have a small spread. There are however some, for e.g. the 28th and 29th, with numerous collocative pixels, that show a higher spread. No clear time evolution of the differences could unfortunately be identified.

Line 388: "a heights" should be "heights" or "altitudes".

Agreed, changed.

Lines 403-405: Perhaps briefly explain the physical processes that cause the bias in TROPOMI retrievals? I assume retrievals are possible with thin clouds above or below the volcanic plume, but only possible with bright clouds below the plume?

Our results highlight that there is added value in study scenes characterized by complexity. We expect that in case of sufficiently ash amount, the cloud height data products provide accurate volcanic ash heights. In case of semi-transparent volcanic ash clouds, where the cloud height retrievals become sensitive to other reflective surfaces below transparent volcanic ash clouds, detection of accurate volcanic ash heights is limited. This is very crucial in most of cases included in our analysis.

The low bias of the TROPOMI-CALIPSO comparisons may be expected as CALIPSO observes the top plume height and TROPOMI observes an average plume height when multiple layers are present. The CALIOP instrument furthermore possibly underestimateσ the aerosol layer thickness due to strong attenuation of the lidar signal at the top of the aerosol layer whereas TROPOMI SO2LH product does not suffer from such attenuation.

Of course, all these are on top of the major fact that the ash and $SO_2$ plumes do not always coincide to begin with.

Figure 11: Perhaps indicate in the figure caption that middle panels are for CAMS without assimilating TROPOMI retrievals and lower panels are with assimilation.

Agreed, changed.

---

## Author Response (AR2)

Review of manuscript "Volcanic SO$_2$ Layer Height by TROPOMI/S5P; validation against IASI/MetOp and CALIOP/CALIPSO observations." by Koukouli et al.

We thank both reviewers for her/his positive comments on our work and all the suggestions which improve our work. Please find our replies insert in red.

*- L.466: missing word: 'by' Iness et al. 2022*

Word added.

*- L.496-497: double use of 'km'*

Double "km" deleted.

*- Figures 1 and S1: As explained in the author response (page 3, comment L.261), values below 20 D.U. and 5 km are depicted by the color of the lowest level. To avoid confusion, the color bar in these figures should therefore have the lowest color level depicted as a downward facing triangle.*

Up- and downward facing triangles are added to Figures 1 and S1, as well as in all such types of Figures in the supplement.

*Please add the country to the last affiliation.*

We have added "Italy" to the last affiliation